# The relevance of mid-Holocene Arctic warming to the future

Masakazu Yoshimori[1,2] and Marina Suzuki[3]

[1] Faculty of Environmental Earth Science, Global Institution for Collaborative Research and Education, and Arctic Research Center, Hokkaido University, Sapporo 060-0810, Japan

[2] Atmosphere and Ocean Research Institute, The University of Tokyo, Kashiwa 277-8568, Japan

[3] Graduate School of Environmental Science, Hokkaido University, Sapporo 060-0810, Japan

*Correspondence to*: Masakazu Yoshimori (masakazu@aori.u-tokyo.ac.jp)

**Abstract.** There remain substantial uncertainties in future projections of Arctic climate change. There is a potential to constrain these uncertainties using a combination of paleoclimate simulations and proxy data, but such a constraint must be accompanied by physical understanding on the connection between past and future simulations. Here, we examine the relevance of an Arctic warming mechanism in the MH to the future with emphasis on process understanding. We conducted a surface energy balance analysis on 10 atmosphere and ocean general circulation models under the MH and future RCP4.5-scenario forcings. It is found that many of the dominant processes that amplify Arctic warming over the ocean from late autumn to early winter are common between the two periods, despite the difference in the source of the forcing (insolation vs. greenhouse gases). The positive albedo feedback in summer results in an increase in oceanic heat release in the colder season when the atmospheric stratification is strong, and an increased greenhouse effect from clouds helps amplify the warming during the season with small insolation. The seasonal progress was elucidated by the decomposition of the factors associated with sea surface temperature, ice concentration, and ice surface temperature changes. We also quantified the contribution of individual components to the inter-model variance in the surface temperature changes. The downward clear-sky longwave radiation is one of major contributors to the model spread throughout the year. Other controlling terms for the model spread vary with the season, but they are similar between the MH and the future in each season. This result suggests that the MH Arctic change may not be analogous to the future in some seasons when the temperature response differs, but it is still useful to constrain the model spread in the future Arctic projection. The cross-model correlation suggests that the feedbacks in preceding seasons should not be overlooked when determining constraints, particularly summer sea ice cover for the constraint of autumn-winter surface temperature response.

## 1 Introduction

The magnitude of climate change has been shown to be larger at high latitudes with paleoclimate evidence (Masson-Delmotte et al. 2013; Masson-Delmotte et al. 2006) and climate model equilibrium simulations (Manabe and Wetherald 1975; Stouffer and Manabe 1999). The Arctic is currently experiencing a more rapid warming than the rest of the world (Screen and Simmonds 2010; Serreze and Barry 2011), and this Arctic amplification is expected to continue at least until the

end of this century (Collins et al. 2013; Laîné et al. 2016). A much slower rate of warming occurs in the Southern Ocean primarily due to oceanic processes (Armour et al. 2016) although it is possible that stratospheric ozone change and cloud feedback play additional roles (Marshall et al. 2014; Yoshimori et al. 2017). A substantial part of the uncertainty in the future Arctic warming projections is attributed to the differences among numerical models (Hodson et al. 2013). In addition, the projected range of future Arctic warming within each RCP scenario is much larger than that for the global mean. For example, the 90% confidence interval for the annual mean surface air temperature (SAT) change from the late 20th century to the late 21st century for the Arctic mean (67.5−90°N) is estimated as 1.6−6.9 °C, while that for the global mean is 1.1−2.6 °C under the RCP4.5 scenario (Collins et al. 2013).

It is often assumed that the study of paleoclimate, particularly of warm periods, is useful for understanding future climate change projections. It is, however, nontrivial to demonstrate the relation between these two different periods. Earlier studies discussed whether past climate can be used as an analogue for the future and refuted the use of past warm periods as an analogue (Crowley 1990; Mitchell 1990). A relatively large number of studies have been conducted on the link between the past, including the last glacial maximum (LGM), and the future in the context of climate sensitivity based on processes and statistical correlation (Crucifix 2006; Hargreaves and Annan 2009; Hargreaves et al. 2007; Hargreaves et al. 2012; Yoshimori et al. 2009; Yoshimori et al. 2011). More recently, broader applications of the relation between paleo and future climate were summarized by Schmidt et al. (2014) who demonstrated the potential to constrain uncertainties using both paleoclimate simulations and proxy data. Indeed, they found a weak statistical inter-model correlation between the sea ice changes in the mid-Holocene (MH) and in future projections (RCP8.5 scenario) relative to the modern period. Such an "emergent constraint" provides a powerful tool to directly reduce the range of uncertainty, provided that the necessary paleoenvironmental information is available. We note that Hargreaves and Annan (2009) also found statistically significant correlations between the mid to high northern latitude temperature for the MH and an elevated $CO_2$ scenario ($2\times CO_2$). The mechanism behind these emergent relations, however, remains unclear.

The purpose of the current study is to investigate commonalities and differences in the Arctic warming mechanisms in the past (MH) and future, and to discuss the relevance of Arctic warming in the MH for understanding future warming based on physical processes. We aim to obtain insight into the feasibility of constraining uncertainty in future climate change projections using paleoclimate data. It is not, however, the purpose of the current study to derive a specific emergent constraint. The MH was chosen because proxy records suggest this period had a warmer Arctic state relative to the pre-industrial period, and multi-model simulation data are available from the Coupled Model Intercomparison Project (CMIP5) data archive (https://cmip.llnl.gov/cmip5/).

The data, models, and experiments are briefly explained in the next section. Analysis methods for diagnosing factors contributing to the surface temperature change in each model and to the inter-model differences are described in Sect. 3. Results are presented in Sect. 4, followed by discussion and conclusion in Sects. 5 and 6, respectively.

## 2 Climate models, experiments, and proxy data

The main analysis in the current study relies on the multi-model simulation data available from the CMIP5 data archive. The preindustrial control (c.a., 1850 C.E.), historical (c.a., 1850-2005 C.E.), and RCP4.5 scenario (2006-2100 C.E.) simulations were designed and coordinated by the CMIP5 project (Taylor et al. 2012). The MH simulation was designed and coordinated by the Paleoclimate Modelling Intercomparison Project (PMIP3) (Braconnot et al. 2012), and later endorsed and archived as part of CMIP5. The MH aims to simulate the climate of approximately 6000 years ago, and the PMIP3 forcing differs only in the earth's orbital configuration (obliquity, seasonal timing of precession, and eccentricity, Table 1) compared to the preindustrial simulations. The difference between the MH and preindustrial (PI) simulations (hereafter, ΔMH) and the difference between the RCP4.5 and historical (HIST) simulations (hereafter, ΔRCP4.5) are compared throughout the paper. For the MH and PI simulations, we use monthly climatological data averaged over periods longer than a century, which were already available. The climatological data are constructed from monthly time series if these data are unavailable from the CMIP5 dataset (Table S1). The 20-year averages for 1980−1999 are used from the HIST simulations and those for 2080−2099 are used from the RCP4.5 simulations, so that ΔRCP4.5 represents the climate change for the entire 21$^{st}$ century, as in Laîné et al. (2016). We use 10 models that produced data for all four experiments (Table 2), and we analyze one simulation run (r1i1p1) for each model and each experiment. Prior to the analysis, all model output data are interpolated onto the T42 Gaussian grid (nominally 2.8°×2.8°) as in Laîné et al. (2016). A common land mask is constructed in such a way that a grid point is judged as ocean if more than 50% of models (that have fractional land cover data) indicate the grid point as ocean. The same procedure is used for the ocean mask, and consequently a small number of grid points are classified as neither ocean nor land.

The simulated ΔMH is compared with temperature reconstructions based on proxy data. Sundqvist et al. (2010) compiled such a dataset primarily based on pollen and chironomids records. The oxygen isotope ratio from ice cores and borehole temperature are also used for the Greenland temperature. Another dataset is compiled by Bartlein et al. (2011) based on pollen records. We use the extended dataset of Bartlein et al. (2011) for the annual mean, which includes additional data from Schmittner et al. (2011) and Shakun et al. (2012) as in Harrison et al. (2014) and is available from the PMIP3 web site (https://pmip3.lsce.ipsl.fr/). The model ensemble mean data are further interpolated onto 2°×2° grids for comparison with Bartlein et al. (2011).

## 3 Analysis method

### 3.1 Surface energy balance and partial temperature changes

Processes contributing to the surface temperature difference between two experiments are evaluated based on the surface energy balance equation. The basic formulation follows Lu and Cai (2010). The surface energy balance equation for a reference climate is given by

$$(1 - \alpha)S + F - R - H - L - Q = 0 \tag{1}$$

where $S = S^{clr} + S^{cld}$ and $F = F^{clr} + F^{cld}$ are the downward shortwave (SW) and longwave (LW) radiation at the surface, respectively, with the superscripts, "clr" and "cld", denoting the clear-sky and cloud (total-sky – clear-sky) radiative effects, respectively. The upward LW radiation is given by the Stefan-Boltzmann law, $R = \sigma T_s^4$, where $\sigma$ is the Stefan-Boltzmann constant and $T_s$ is the surface temperature. The surface emissivity is assumed to be one. $H$ and $L$ are the net upward sensible and latent heat fluxes, respectively, and $Q$ represents the net downward surface energy flux including the latent heat consumed by snow/ice melting. In the ocean, $Q$ is stored locally or transported. For the difference ($\Delta$) between the two experiments, Eq. (1) becomes

$$4\sigma T_s^3 \Delta T_s = \begin{bmatrix} -\Delta\alpha S - \Delta\alpha\Delta S + (1-\alpha)\Delta S^{clr} + (1-\alpha)\Delta S^{cld} \\ +\Delta F^{clr} + \Delta F^{cld} - \Delta H - \Delta L - \Delta Q \end{bmatrix} \equiv \sum_j \Delta R_j \tag{2}$$

where $\Delta R_j$ represents the individual energy terms.

The Stefan-Boltzmann law implies that a larger surface warming ($\Delta T_s$) is required to balance the same amount of energy flux anomaly ($\Delta R$) by emitting LW radiation at a colder background temperature ($T_s$). Laîné et al. (2016) called this effect the "surface warming sensitivity", whose importance for the Arctic amplification has been pointed out in multiple studies (Laîné et al. 2016; Laîné et al. 2009; Ohmura 1984, 2012; Pithan and Mauritsen 2014). The warming sensitivity and other energy flux terms may be converted to the same temperature scale (partial surface temperature changes) by

$$\Delta T_s = \overline{\left(\frac{\partial T_s}{\partial R}\right)} \sum_j \Delta R_j{}' + \left(\frac{\partial T_s}{\partial R}\right)' \sum_j \overline{\Delta R_j} + \left(\frac{\partial T_s}{\partial R}\right)' \sum_j \Delta R_j{}' \tag{3}$$

where overbars and dashes represent the global mean and deviations from the global mean (local anomaly), respectively, and

$$\frac{\partial T_s}{\partial R} = \frac{1}{4\sigma T_s^3} \tag{4}$$

Equation (3) enables the quantification of the effect of a colder winter Arctic requiring more warming to balance the anomalous surface energy flux on the same partial temperature change scale as other components. The left side of Eq. (3) is the simulated surface temperature change. The first, second, and third terms on the right side of Eq. (3) represent local feedbacks evaluated with the global mean warming sensitivity, global mean feedbacks with the local warming sensitivity, and local feedbacks with the local warming sensitivity, respectively. Note that previous studies used the tropical mean in place of the global mean (Laîné et al. 2016; Pithan and Mauritsen 2014). In Sects. 4.3 and 4.5, each component of the first term is evaluated separately, and the second and third terms are evaluated collectively as the "S-B" effect and "synergy" effect, respectively (Table 3). Accordingly, the surface temperature change formulated by Eqs. (2) and (3) can be written in a more explicit form as

$$\Delta T_s = \overline{\left(\frac{\partial T_s}{\partial R}\right)} \left\{ \begin{matrix} -(\Delta\alpha S)' - (\Delta\alpha\Delta S)' + [(1-\alpha)\Delta S^{clr}]' + [(1-\alpha)\Delta S^{cld}]' \\ +(\Delta F^{clr})' + (\Delta F^{cld})' - (\Delta H)' - (\Delta L)' - (\Delta Q)' \end{matrix} \right\} + \left(\frac{\partial T_s}{\partial R}\right)' \sum_j \overline{\Delta R_j} + \left(\frac{\partial T_s}{\partial R}\right)' \sum_j \Delta R_j{}'$$

$$\equiv (\text{alb}) + (\text{alb*clr\_sw}) + (\text{clr\_sw}) + (\text{cld\_sw}) + (\text{clr\_lw}) + (\text{cld\_lw}) + (\text{sens}) + (\text{evap}) + (\text{surface})$$

$$+(\text{S-B}) + (\text{synergy}) \tag{5}$$

Here $\alpha$ is computed from the ratio of upward to downward SW radiations at the surface; $S$, $F$, $H$, and $L$ are taken directly from the model output; and $Q$ is computed as a residual of surface heat fluxes (net radiation, sensible heat, and latent heat fluxes). $\sum_j \Delta R_j$ is computed by summing changes in all surface energy flux terms after either averaged globally for overbars or the global average is subtracted for dashes. We use the average of $T_s$ from the paired experiments (PI and MH, or HIST

and RCP4.5) to calculate $\partial R / \partial T_s$. Although using the average of two experiments or a single experiment for this term has little impact on the results of the current study, we found that the average provided better agreement between the two sides of Eq. (5) for larger perturbations such as a quadrupling of the $CO_2$ experiment. The diagnosis is made for each grid point and each month. All models are used in this analysis. We note that direct comparisons between different forcing simulations are possible as there is no change in the land-sea mask among the simulations.

**3.2 Interpretation of surface temperature change at partially ice-covered ocean grid points**

The surface temperature archived in the CMIP5 dataset represents the grid-mean "skin" temperature. At the fractionally ice-covered ocean grid points, this variable is a mixture of the sea surface temperature (SST) and ice surface temperature. We assume that the surface temperature $T_s$ at each grid point is reconstructed by

$$T_s = (1 - A)T_o + AT_i \tag{6}$$

where $T_o$ and $T_i$ are the SST and ice surface temperature, respectively, and $A$ is the ice concentration. The factors contributing to the surface temperature difference for the paired experiments are then diagnosed by

$$\Delta T_s = (1 - A)\Delta T_o + A\Delta T_i + (T_i - T_o)\Delta A. \tag{7}$$

The first and second terms on the right side represent the effect of SST and ice surface temperature changes, respectively. The last term on the right side represents the effect of the ice concentration change, which is weighted by the surface

temperature difference between ice and water: the reduction of sea ice cover ($\Delta A<0$) and the exposure of the warmer ocean surface to the atmosphere ($T_i - T_o < 0$) lead to an increase in the grid-mean surface temperature ($\Delta T_s$). In the current analysis, $T_o$, $T_i$, and $A$ are obtained from the average of paired experiments. We use $T_o$ in place of $T_i$ for ice-free ocean grids. Only five models (bcc-csm-1, CCSM4, CNRM-CM5, IPSL-CM5A-LR, and MRI-CGCM3) are used for this analysis due to the availability of the required variables, and the consistency of the analysis is verified by agreement between the left and

right sides of Eq. (7). The diagnosis is made for each grid point and each month.

**3.3 Factors responsible for the model spread**

The fractional contribution of individual partial surface temperature changes (or feedbacks in other words) to the inter-model spread of the simulated surface temperature change is given by

$$V_j = \sum_{k=1}^{n} \frac{(\Delta T_{j,k} - \overline{\Delta T}_j)(\Delta T_k - \overline{\Delta T})}{\sigma^2(n-1)} \times 100 \ [\%] \tag{8}$$

where $V_j$ is the fractional contribution and $\Delta T$ is the surface temperature change (the subscript "s" in $\Delta T_s$ is omitted here). The subscripts $j$ and $k$ denote indices for feedbacks ($j$th feedback) and models ($k$th model out of $n$ models), respectively.

The overbars denote the average over the feedbacks ($\overline{\Delta T_j}$), or over both the feedbacks and models ($\overline{\Delta T}$). $\sigma$ is the inter-model standard deviation of the total surface temperature change. The numerator represents the product of the model spread for each feedback and the model spread for the total feedback, while the denominator represents the ensemble variance of the total feedback. Here, the key points are: 1) $V_j$ accounts for 100% of the surface temperature change when summed over the feedbacks; 2) positive $V_j$ means that the $j$th feedback amplifies the model spread, while negative $V_j$ means that it suppresses the model spread. We note that the same formula was used in Yoshimori et al. (2011) and the references therein. The statistical significance of the fractional contribution is tested using the Monte Carlo method by randomly shuffling the model index ($k$) $10^5$ times. The null hypothesis is that the $V_j$ neither amplify nor suppress the model spread. When the original $V_j$ is outside the range of the 5−95th percentile of $V_j$ resulting from the shuffling, it is considered significant. The diagnosis is made separately for ocean and land averages in the Arctic region (north of 60°N). All models are used for this analysis.

## 4 Results

### 4.1 Simulated surface air temperature response

Figure 1 shows the ensemble mean of the annual mean SAT response for ΔMH and ΔRCP4.5. In both cases, the warming in the polar regions is larger than for the rest of the world, particularly in the Arctic. The Arctic mean response is 0.4 °C and 3.9 °C for ΔMH and ΔRCP4.5, respectively, whereas the global mean response is −0.2 °C and 1.9 °C for ΔMH and ΔRCP4.5, respectively (see Table 2 for individual models). This feature reflects the so-called Arctic warming amplification in ΔRCP4.5. The warming at high latitudes and cooling at low latitudes in ΔMH are consistent with the annual mean insolation anomaly caused by the obliquity difference. From this figure it is unclear whether the Arctic warming in ΔMH is due to forcing and/or feedbacks.

Figures 2a and 2b show the seasonal progress of the effective radiative forcing (ERF) for ΔMH and ΔRCP4.5, respectively. The ERF is the top-of-the-atmosphere (TOA) radiation change induced by the forcing constituents and is computed here using the atmospheric GCM (MIROC4m) of Yoshimori et al. (2018), with prescribed climatological SST and sea ice distribution. The ERF for ΔMH was computed by applying the PI and MH insolation to the AGCM separately with other boundary conditions held fixed. The TOA net radiation in the MH was averaged for 20 years after a 10-year spin-up and the difference from the PI was taken as ΔMH ERF. The ERF for ΔRCP4.5 was drawn using the data from Yoshimori et al. (2018) in which the time-varying historical and RCP4.5 forcing were applied continuously to the AGCM with other boundary conditions held fixed. The 3-ensemble-member mean of the differences between the 2080–2099 and 1980-1999 averages was taken as ΔRCP4.5 ERF. While this so-called Hansen-style method (Flato et al. 2013; Hansen et al. 2005) is one of the standard procedures for calculating future scenario forcing, e.g., ΔRCP4.5, it is uncommon in paleoclimate applications. With this method, the ERF includes both rapid stratospheric and tropospheric adjustments as well as the land surface response to the instantaneous radiative forcing. Although the land surface response should not be considered as a

forcing, we present the ERF to facilitate a consistent comparison between different perturbation experiments. As a supplementary reference, another measure of radiative forcing evaluated by $\Delta S(1 - \alpha_p)$ is presented for $\Delta$MH in Fig. S1. Here, $\Delta S$ is the insolation anomaly and $\alpha_p$ is the pre-industrial planetary albedo. The $\Delta$MH forcing patterns in both Fig. 2a and Fig. S1 are qualitatively similar to the familiar insolation anomaly $\Delta S$ (e.g., Hewitt and Mitchell 1996; Ohgaito and Abe-Ouchi 2007): an increase and a decrease in summer and autumn, respectively, in the Northern Hemisphere, and an increase and a decrease in spring and summer, respectively, in the Southern Hemisphere. For the Arctic average (> 60 °N), the peak positive ERF of about 19.9 W m$^{-2}$ occurs in July and the peak negative ERF of about −4.8 W m$^{-2}$ occurs in September. The $\Delta$RCP4.5 ERF is, in contrast, spatially and seasonally more homogeneous with an annual mean of about 3.0 W m$^{-2}$ for the Arctic region. Figures 2c and 2d show the ensemble mean of the seasonal progress of SAT changes for $\Delta$MH and $\Delta$RCP4.5, respectively. A common and striking feature is that the maximum Arctic warming occurs in autumn (though the magnitude differs substantially) when the ERF is negative or weakly positive. This result suggests that feedbacks play an important role in shaping the seasonality of the Arctic warming for both $\Delta$MH and $\Delta$RCP4.5. This interpretation is in line with Zhang et al. (2010) for $\Delta$MH and Laîné et al. (2016) for $\Delta$RCP4.5.

Figure 3 shows SAT changes over the land and ocean for individual models. The seasonality of the SAT change over land is distinct between $\Delta$MH and $\Delta$RCP4.5, but there are some similarities over the ocean: the warming is modest in summer and largest in autumn. Significantly, the model spread over the ocean is also larger in autumn than in summer. The maximum land warming in summer for $\Delta$MH corresponds to the maximum local insolation anomaly, and it thus may appear that the SAT warming over land is not related to the SAT warming over the ocean. However, there are strong cross-model correlations at the 5% statistical significance level (Student's two-tailed $t$-test) between the Arctic land and ocean for the October-November-December (OND) mean as well as for the annual mean (0.95 for OND and 0.94 for the annual mean). The statistically significant cross-model correlations at the 5% level also exist for $\Delta$RCP4.5 (0.92 for OND and 0.89 for the annual mean). In addition, the inter-model variance of the Arctic-mean SAT anomaly is larger over the ocean than over land. Although the available surface temperature proxy data for the mid-Holocene Arctic are more abundant on land than over the ocean (Bartlein et al. 2011; Sundqvist et al. 2010), it is useful to focus our analysis on the oceanic region, which has a larger response, and to explore which processes are responsible for the model difference there. We note that there is no statistically significant correlation at the 5% significance level between $\Delta$MH and $\Delta$RCP4.5 for either the OND or annual means (for both the Arctic ocean and land).

**4.2 Comparison with proxy data**

Figure 4 shows the ensemble mean of the simulated $\Delta$MH annual mean, July, and January SAT anomalies superimposed with the reconstructed SAT anomaly at proxy sites taken from Sundqvist et al. (2010). We note that a detailed comparison with earlier PMIP1 and PMIP2 simulations was given by Zhang et al. (2010). There is substantial disagreement between the model and the reconstruction: the warming indicated by the reconstruction is not captured by the model mean in January as

well as in the annual mean. The discrepancies are on the order of a few degrees. Although better agreement is seen in July, the simulated warming is overestimated at some North American sites. O'Ishi and Abe-Ouchi (2011) reported that the model-data discrepancy improved substantially when the interaction between the MH climate change and vegetation distribution change is included in one model although the improvement is somewhat limited in other models (Zhang et al. 2010).

Unfortunately, none of the models analyzed in the current study include this dynamic vegetation feedback. Comparisons of the model ensemble mean with Bartlein et al. (2011) for the ΔMH annual mean, warmest month, and coldest month are shown in Fig. S2. We note that a more comprehensive comparison with PMIP2 and PMIP3 simulations was presented in Harrison et al. (2014). Again, the model-data discrepancy is large although the qualitative tendencies of the warming in parts of Scandinavia appear in both. While these limitations need to be kept in mind, they do not reduce the significance of the

following results on the understanding of the Arctic warming process. As stated in the introduction, it is not the purpose here to derive a specific emergent constraint using these proxy data, as such a study requires a rigorous statistical approach in parallel to the mechanism understanding and appropriate proxy searches, and is beyond the scope of this article.

## 4.3 Partial temperature changes

Figure 5 shows the contribution of individual energy flux components to the surface temperature change (partial $T_s$

changes) in the Arctic ocean diagnosed by the feedback analysis described in Sect. 3.1. As expected, the simulated $T_s$ changes (black polygonal solid lines) are reproduced by the sum of the individual contributions (blue polygonal dashed lines), indicating that the decomposition is useful.

In spring (March-April-May), the total surface temperature change is negative for the case of ΔMH, whereas it is positive for ΔRCP4.5. Therefore, there is no analogy in the response between the two cases. While the synergy effect of local Arctic

feedbacks and local warming sensitivity (synergy) slightly contributes to the warming in both cases, the contributions from the downward clear-sky LW radiation components (clr_lw) have opposite signs between the two cases. The albedo feedback (alb) exhibits a relatively large warming effect for ΔRCP4.5, accompanied by cooling due to the surface effect in late spring (net surface heat flux component, or equivalently ocean heat storage and dynamics components). On the other hand, the surface effect is positive for ΔMH, and is accompanied by anomalous turbulent heat fluxes from the ocean to the atmosphere

(evap and sens).

In summer (June-July-August), the total surface temperature change is positive but small for both ΔMH and ΔRCP4.5. The albedo feedback is distinctly positive for both cases. An even larger clear-sky SW radiation component (clr_sw) contributes to the additional warming for the case of ΔMH, which is largely driven by the astronomical forcing but it plays little role in ΔRCP4.5. The increased SW radiation reaching the sea surface through the albedo feedback and/or increased

seasonal insolation is counteracted by the increased net surface heat flux component, implying that the extra energy is likely stored in the form of ocean heat content. The net result is a small surface warming in summer. It is a common feature of both ΔMH and ΔRCP4.5 that the SW cloud radiative effect (cld_sw) weakens warming by the albedo feedback. This cancelling

role of clouds in the warm season is consistent with previous studies using future climate projections (Crook et al. 2011; Laîné et al. 2016; Lu and Cai 2009). In both cases, the downward clear-sky LW radiation component plays a substantial role in warming the surface (except for ΔMH in June).

From September to January, the total surface temperature change is larger than in other seasons for both ΔMH and ΔRCP4.5. From September to November, the clear-sky SW radiation component associated with the astronomical forcing contributes to the surface cooling for ΔMH, which is absent for ΔRCP4.5. From October to January for both ΔMH and ΔRCP4.5, the positive surface effect is counteracted by the negative surface turbulent flux components, indicating that the heat is released from the ocean to the atmosphere in the form of latent and sensible heat fluxes. It is, however, unclear how the heat release to the atmosphere leads to the surface warming (or, more precisely, grid-mean skin temperature rise). This point is discussed in the next subsection in detail. It is a common feature of both ΔMH and ΔRCP4.5 that the LW cloud radiative effect (cld_lw) helps warming by the surface effect. This amplifying role of clouds in the cold season is consistent with previous studies using future climate projections (Laîné et al. 2016; Yoshimori et al. 2014). The general increase of cloud cover in autumn to winter for both ΔMH and ΔRCP4.5 is consistent with the enhanced greenhouse effect of clouds (Figs. 6a and 6c).

Throughout the year, the downward clear-sky LW radiation component exhibits a large contribution and follows the shape of the seasonal progress of the total response for ΔRCP4.5. This component is, however, not large in winter (and June) for ΔMH. This term includes the effect of air temperature and specific humidity changes (and also the radiative forcing of greenhouse gases for the case of ΔRCP4.5), and is qualitatively consistent with changes in both variables (Figs. 7a and 7c, and Figs. 8a and 8c). Obtaining a clear physical interpretation of its role in the surface temperature change is difficult because the primary component of clear-sky LW radiation is emitted from the atmospheric layer near the surface (Ohmura 2001) where the temperature is tightly coupled with the surface, thus obscuring the causality. Nevertheless, the importance of this component has been reported in previous studies (Pithan and Mauritsen 2014; Sejas and Cai 2016). The positive local feedbacks in the cold season with a larger local warming sensitivity make the synergy term an important contributor to the total response for both ΔMH and ΔRCP4.5, as found by Laîné et al. (2016) in future climate projections. For completeness, the same analysis for the land surface temperature is shown in Fig. S3.

**4.4 Interpretation of surface temperature change in partially ice-covered ocean grids**

Figure 9 shows the surface temperature change (left side of Eq. (7), $\Delta T_s$) and the individual contributions of surface conditions (the individual terms on the right side of Eq. (7)). The surface air temperature change ($\Delta T_a$) is also plotted for reference. The seasonal progress of $\Delta T_a$ closely follows that of $\Delta T_s$, suggesting the importance of understanding the grid-mean surface temperature change. The surface and surface air temperature changes have maximum values of 3.3 and 2.9 °C, respectively, in October for ΔMH. They have maximum values of 10.2 and 9.3 °C in November for ΔRCP4.5. The figure indicates that the large increase in grid-mean surface temperature during winter is largely due to the ice surface temperature

increase when the SST anomaly decreases seasonally through oceanic heat release after its peak value (Figs. 10a and 10c). The contribution from ice temperature change has a maximum value of 2.2 °C in October for ΔMH and 6.8 °C in November for ΔRCP4.5. The contribution from SST change has a maximum value of 0.7 °C for ΔMH and 2.0 °C for Δ RCP4.5, both in August. The magnitude of the SST anomaly effect on the grid-mean surface temperature change is small as the SST change

itself is small because SST is fixed at the melting point where sea ice is present and due to the large heat capacity of sea water. The reduction of sea ice cover makes an important contribution to the grid-mean surface temperature increase during autumn. Its peak contribution does not, however, coincide with the timing of the maximum ice concentration anomaly ($\Delta A$, Figs. 11a and 11c) as the effect is weighted by the surface temperature difference between the sea ice and ocean ($T_i - T_o$). The interpretation of the results of the feedback analysis in the previous section is that the oceanic heat release in the cold

season represented by the positive net surface heat flux term in Fig. 5 contributes to the surface air temperature rise and subsequent ice (and grid-mean) surface temperature rise. This diagnosis is simple but reveals a chain of processes whose temporal links are less clear from the conventional analysis on surface energy balance alone.

### 4.5 Factors for the inter-model difference in surface temperature changes

Figure 12 shows the fractional contribution of the partial surface temperature changes to the model spread in the total

surface temperature changes. The average is taken for the Arctic ocean areas, and positive or negative values indicate factors increasing or reducing the model differences, respectively. In the following, individual components whose contributions are either small or inconsistent between the ΔMH and ΔRCP4.5 cases are not discussed, after considering the statistical significance.

In spring (Fig. 12a), large contributions to the model spread are made by the albedo feedback (alb) and the downward

clear-sky LW radiation component (clr_lw) for both ΔMH and ΔRCP4.5. Each of these factors contributes to more than 50% of the model spread. LW cloud feedback (cld_lw) and the synergy effect of local Arctic feedbacks and local surface warming sensitivity (synergy) also contribute to the model spread, but to a lesser degree. In contrast, the turbulent heat flux components (evap and sens) as well as the cloud SW radiation component (cld_sw) tend to suppress the model spread.

In summer (Fig. 12b), the albedo feedback (alb) exhibits by far the largest (more than 170%) contribution to the model

spread for both ΔMH and ΔRCP4.5. Note that the vertical scale in Fig. 12b is enlarged three-fold compared to other plots. As in spring, the downward clear-sky LW radiation component also contributes to more than 50% of the model spread. The surface effect (net surface heat flux component, or equivalently ocean heat storage and dynamics components) substantially suppresses the model spread for ΔMH, but it is insignificant for ΔRCP4.5.

In autumn and winter (Figs. 12c and 12d), the downward clear-sky LW radiation component, LW cloud feedback, and

surface effect contribute to the model spread, whereas the turbulent heat flux components tend to suppress it for both ΔMH and ΔRCP4.5. As the oceanic heat content is reduced in these seasons through latent and sensible heat fluxes, it is understandable that these two terms have opposite sign to the surface effect, similar to how the albedo feedback and surface effect have opposite signs in summer. The surface effect contributes to more than 40% of the model spread in autumn and

more than 50% in winter for both ΔMH and ΔRCP4.5. In contrast to spring and summer, the contribution by the albedo feedback is small in autumn and winter.

The downward clear-sky LW radiation consistently exhibits a large positive contribution (more than 50%) in all seasons for both ΔMH and ΔRCP4.5. The clear-sky LW radiation is often dominant for ΔMH and ΔRCP4.5 even in spring when the ensemble mean shows surface cooling in ΔMH and warming in ΔRCP4.5. It is also one of major contributors to the model spread even in winter when there is little contribution from the clear-sky LW radiation to the ensemble mean response of ΔMH. The large contribution of this term to the model spread is somewhat expected because this radiative flux largely reflects the surface air temperature, which is thermally coupled with the surface temperature as shown in the previous section. This term, however, also includes the effect of water vapor and lapse-rate changes, whose quantitative contributions are not evaluated separately here. The model variances of the air temperature change are concentrated near the surface in non-summer season (Figs. 7b and 7d), and those of the specific humidity change are large in non-spring season (Figs. 8b and 8d). The relative contribution of air temperature and water vapor to the clear-sky LW radiation may thus vary with the season.

It is also important to point out that the LW cloud feedback contributes positively to the model spread in almost all seasons for both ΔMH and ΔRCP4.5. While the inter-model variability in cloud cover peaks in summer for ΔMH and late autumn for ΔRCP4.5 (Figs. 6b and 6d), the result suggests that the correct representation of LW cloud feedback is important throughout the year. It is important to recognize that the cloud response is not, however, independent of other feedbacks such as sea ice cover, water vapor, lapse rate, large-scale condensation, and convection (cf. Abe et al. 2016; Yoshimori et al. 2017). It is also important to notice that the synergy term contributes positively to the model spread. As the surface warming sensitivity depends on the background temperature, this result may suggest that the differences in the reference surface temperature, i.e., model bias, has the potential to reduce the simulated model spread. Taken together, attention needs to be paid to the models' representation of surface albedo, turbulent heat fluxes (and thus the atmospheric stratification including inversion), clouds, and temperature bias to reduce the differences in the models' response.

These results suggest that the processes responsible for the model spread may depend on the season. While the albedo feedback shows only a small contribution to the autumn-winter model spread, this result does not mean that the summer albedo feedback is irrelevant to the model spread in autumn-winter, however. As the reduction of sea ice cover is considered to enhance the oceanic heat uptake through the enhanced albedo feedback, and the reduction of sea ice cover is also considered to enhance the oceanic heat release through the reduced thermal insulating effect, a chain of processes is expected. The model variances of the sea ice concentration change are large from late summer to early autumn with peaks in September-October for both ΔMH and ΔRCP4.5 (Figs. 11b and 11d), and the model variances of the ocean heat content change are also large in late summer to early autumn, although the peaks occur slightly earlier (Figs. 10b and 10d). These results are not sufficient to prove the existence of inter-seasonal linkage, but they are consistent with its existence. We calculate cross-model correlations between the summer albedo feedback and October-November-December (OND) feedbacks. The correlations of the summer albedo feedback are 0.72 (ΔMH) and 0.60 (ΔRCP4.5) with the OND surface effect, 0.66 (ΔMH) and 0.69 (ΔRCP4.5) with the OND LW cloud feedback, and 0.85 (ΔMH) and 0.87 (ΔRCP4.5) with the

OND surface temperature response (i.e., sum of all feedbacks). These values are statistically significant at the 5% level according to a Student's two-tailed t-test. The significant correlations with the surface effect and with the cloud greenhouse effect are consistent with the chain of processes discussed in Sect. 4.4 and in previous studies (e.g., Abe et al. 2016). Therefore, the model spread in the OND surface temperature response is closely related to the summer sea ice distribution, indicating that feedbacks in preceding seasons should not be overlooked. The recent sensitivity experiment with a single model by Park et al. (2018) demonstrates the dominant influence of sea ice albedo feedback on the MH Arctic winter and annual mean warmings. For completeness, the same analysis for the land surface temperature is shown in Fig. S4.

## 5 Discussions

While the ensemble mean surface temperature response over the Arctic ocean shows a consistent warming trend from summer to autumn for both ΔMH and ΔRCP4.5, the temperature anomaly in spring is neutral or negative for ΔMH and positive for ΔRCP4.5. Although the source of the peak negative anomaly occurring in April for ΔMH is unclear without dedicated numerical experiments, the zonal mean patterns of ERF and surface air temperature change in Fig. 2 suggest that it may originate from a negative insolation anomaly at lower latitudes. This interpretation is consistent with the downward clear-sky LW radiation contributing to the surface cooling. In addition, the peak mid-tropospheric cooling in spring and warming in summer for ΔMH in Fig. 7a are suggestive of remote influence through atmospheric heat transport. The significant remote influence on the Arctic temperature change has been suggested by previous studies in the context of future climate change (Stuecker et al. 2018; Yoshimori et al. 2017). The opposite signs in the total surface temperature change and also in the partial temperature change by downward clear-sky LW radiation between ΔMH and ΔRCP4.5 do not suggest a strong similarity between MH and future Arctic response in spring. While the ensemble mean surface temperature response over the Arctic ocean shows relatively small warming in summer for both ΔMH and ΔRCP4.5, they are the downward clear-sky SW radiation for ΔMH and albedo feedback for ΔRCP4.5 that dominate in the partial temperature changes. Nevertheless, the increased absorption of SW radiation by the ocean and increased reflection of SW radiation by clouds occur for both ΔMH and ΔRCP4.5, suggesting that the relevant processes are controlling the Arctic response in summer. The positive partial temperature changes by the surface effect, cloud greenhouse effect, and synergy effect are common in ΔMH and ΔRCP4.5 in autumn. Together with the concurrent largest warming, it is suggested that the MH Arctic warming in this season is strongly relevant to the future Arctic warming. While the contribution from downward clear-sky LW radiation to the partial temperature change is large throughout the year for ΔRCP4.5, it plays a role only in some months for ΔMH. As the near-surface air temperature is thermally coupled to the surface temperature as shown in Fig. 9, it was thought that the partial temperature change by downward LW radiation behaves similarly to the total surface temperature change. In the ΔMH, however, the contribution by this component is small in winter. As this term consists of vertically uniform temperature change, lapse rate change, and water vapor change, the different behavior does not immediately mean that the mean tropospheric temperature response is decoupled from the surface. Nevertheless, it is possible that the different behavior

is caused by the remote influence from lower latitudes where insolation is reduced for ΔMH. In any case, this difference may weaken the similarity in the surface temperature response between ΔMH and ΔRCP4.5.

As expected from the magnitude of the influence, the processes found to be important for the warming trend from summer to autumn in ΔMH and ΔRCP4.5 are also primarily responsible for the model spread in these seasons. What is interesting is that the processes contributing to the model spread in other seasons are relatively similar between ΔMH and ΔRCP4.5 even when the ensemble mean surface temperature response is very different. The most notable example is spring when cooling occurs in ΔMH and warming occurs in ΔRCP4.5. Such a discordance can occur because the feedback with the largest magnitude is not necessarily the feedback with the most uncertainty. In the global mean radiative feedback analogy, for example, Planck and water vapor feedbacks have large magnitude but the response to the smaller SW cloud feedback is thought to contain the most uncertainty. In spring, the albedo feedback and downward clear-sky LW radiation are the major contributors to the model spread. As discussed in the above, the temperature response in ΔMH is not highly similar to the future Arctic response in this season. Nevertheless, the model spread occurs through similar feedback processes. This result suggests that if the models are constrained by ΔMH proxy reconstruction in this season, there is a potential that the constraint may affect the future Arctic projection in the same season even though the response is not alike. In this sense, ΔMH Arctic change is useful for constraining future Arctic projection in all seasons. However, the confirmation of this statement requires a rigorous statistical analysis.

In the current analysis, the target variable of interest is surface temperature change, and an emphasis was made on atmospheric feedbacks. Previous studies reported that many important feedbacks also reside in the interaction of sea ice and ocean (Goosse et al. 2018). For example, sea ice grows faster when it is thin and this feedback works to counter warming. While sea ice related terms such as albedo feedback (a function of ice cover among others) and heat release from the ocean (a function of ice thickness among others) are diagnosed, the ice thickness feedback itself was not quantified in the current study. Such a diagnosis would require an energy budget analysis for sea ice and probably for the mixed-layer ocean as well, and it is worth further investigation in the future.

Recently, Hu et al. (2017) argued that "the global warming projection spread...is inherited from the diversity in the control climate state." They also pointed out a possibility that the diversity of feedbacks can arise from the same control climate state which may be constructed from the compensation of different processes. We add to these points that there may be a systematic bias or uncertainty due to common, missing feedbacks in many climate models that do not appear as the model spread. The paleoclimate has the potential to provide a constraint for the future projections in the second and third cases, beyond the emergent constraint. Related to this discussion, there remains an outstanding issue to be explored. O'Ishi and Abe-Ouchi (2011) showed that the vegetation change in response to climate change in both the mid-Holocene and elevated $CO_2$ experiments amplifies the Arctic warming. In particular, the expansion of boreal forest in place of tundra lowers the surface albedo through earlier snow melting and leads to the amplification of continental warming in spring and subsequent maritime warming in winter. None of the models analyzed in the current study include the effect of climate-vegetation

interaction. Therefore, the conclusion of the current study needs to be verified by models with a dynamic vegetation component.

The current study focuses on the mid-Holocene partly because multi-model simulations for this period are easily accessible through the CMIP5 data archive, and the compiled reconstruction dataset is also available. There are, however, other periods that appear to exhibit larger Arctic warming such as the last interglacial (MIS5e), MIS11, and mid-Pliocene (Berger et al. 2016; Dutton et al. 2015; Lunt et al. 2013). These warm periods surely would be useful for expanding the analysis conducted in this study. While the energy balance feedback analysis has been applied to the MH, LGM, and mid-Pliocene (Braconnot and Kageyama 2015; Hill et al. 2014), which are very useful for understanding past climate change, a study focusing on the relevance to the future is encouraged. It should be straightforward to expand the current study to other periods once the multi-model simulations are easily accessible. In addition, the current analysis does not separate the downward LW radiation in the Arctic region into local and remote origins, and thus provides only a local feedback perspective. As the change in orbital configurations redistributes the insolation latitudinally, a significant change in the meridional heat transport is expected. The change in the meridional heat transport by both the atmosphere and ocean in response to the wider variety of orbital configurations is worth further investigation in the future. Furthermore, expanding the current study to cases with more general astronomical forcing (e.g., only considering the effect of the obliquity change or precession change), and to consider the implications for the mechanism for glacial-interglacial cycles (e.g., Abe-Ouchi et al. 2013) may also be valuable.

## 6 Conclusions

The relevance of Arctic warming mechanisms in the MH to the future under the RCP4.5 scenario was investigated. The emphasis was placed on the surface temperature change over the ocean where peak warming occurs nearly in the same season for both periods and the model spread is large. Although the insolation in the Arctic region decreases in autumn for the MH relative to the modern period, the largest MH Arctic warming occurs in autumn. Although the elevated $CO_2$ radiative forcing is rather uniform globally and seasonally, the largest future Arctic warming also occurs almost in the same season as for the MH. Within the limited range of processes investigated, the current study suggests that the dominant processes causing the Arctic warming trend from summer to autumn in the MH and in the future are common: positive albedo feedback in summer (though partially counteracted by the sunshade effect from clouds), the consequent increase in heat release from the ocean to the atmosphere in the colder season when the atmospheric stratification is strong, and an increased greenhouse effect from clouds during the season with small insolation. A chain in the seasonal progress was elucidated by a decomposition into factors associated with SST, ice concentration, and ice surface temperature changes, whose temporal links are less clear from the conventional surface energy balance analysis alone. In addition, the synergy effect of local Arctic feedbacks and local warming sensitivity contributes to the enhanced warming during the cold season for both cases. There are some differences, however. The contribution from the downward clear-sky SW radiation is large positive in

summer and negative in autumn for the MH, but it plays only a minor role in the future. Furthermore, the large contribution from the downward clear-sky LW radiation occurs throughout the year for the future projections, but it is only distinct in April-May and July-October for the MH.

The downward clear-sky LW radiation is one of the major contributors to the model spread for surface temperature changes throughout the year. Although whether this term originates from remote sources or local feedbacks is unclear from the current analysis, the importance of this term is common for the model spread in the MH and the future simulations. The processes found to be important for the warming trend from summer to autumn (albedo feedback, surface effect, cloud greenhouse effect, and synergy effect) are also found to be primarily responsible for the model spread in these seasons. The dominant feedbacks for the model spread depends on the season—albedo feedback for spring and summer, and surface effect for autumn and winter—although the importance of the inter-seasonal linkage of feedbacks is not excluded. Cloud feedbacks are less important for the model spread in summer and a small contribution from downward clear-sky SW radiation is found throughout the year.

The fact that MH Arctic ocean warming is moderate in all seasons except for late autumn to early winter and the model spread is large in the cold season underlines the importance of model validation with proxy reconstruction in the cold season. However, the factors contributing to the model spread are also common between the MH and the future in other seasons, including spring, when opposite signs of temperature response occur. This result suggests that the MH Arctic change may not be directly relevant to the future in some seasons but it is still useful to constrain the future Arctic projection. In this sense, the seasonal evolution of surface temperature response in the MH Arctic is a useful variable. In practice, however, the available constraint would be limited to the cold season when the temperature response over the ocean is well correlated with that over land across models. The significant correlation found between the summer albedo feedback and autumn-winter temperature response across models suggests that feedbacks in preceding seasons should not be overlooked and the sea ice cover may be another useful constraint.

The relevance between past and future climate arises not only from a common forcing to the climate system but also from the feedbacks inherent in the climate system. While basic physical principles do not change with time, it is not trivial that the dominating processes for the climate variations are the same for different climate forcing and response. Therefore, more effort should be made in seeking possible analogues in the dominant physical processes between the past and future climate, rather than in the past forcing. The following points are highlighted from the current study.

- Many of the dominant processes that amplify Arctic warming over the ocean from late autumn to early winter are common between the two periods, despite the difference in the source of the forcing (insolation vs. greenhouse gases).
- A chain of processes responsible for the warming trend from summer to autumn can be elucidated by the decomposition to factors associated with SST, ice concentration, and ice surface temperature changes.
- The downward clear-sky longwave radiation is one of major contributors to the model spread throughout the year. Other controlling terms vary with the season, but they are similar between the MH and the future in each season.

- The MH Arctic change may not be analogous to the future in some seasons when the temperature response differs, but it is still useful to constrain the model spread in the future Arctic projection.

- The significant cross-model correlation found between the summer albedo feedback and autumn-winter surface temperature response in both forcing cases suggests that the feedbacks in preceding seasons, particularly sea ice cover, should not be overlooked when determining constraints.

**Data availability**

The PI, MH, HIST, and RCP4.5 simulation data can be downloaded from the ESGF server (https://esgf-node.llnl.gov/search/cmip5/, last access: 12 March 2019) (ESGF, 2019) as piControl, midHolocene, historical, and rcp45. Temperature reconstructions from proxy data used in Fig. 4 are taken from Table 1a of Sundqvist et al. (2010). Temperature

reconstructions from proxy data used in Fig. S2 can be downloaded from the PMIP3 web site (https://pmip3.lsce.ipsl.fr/, last access: 12 March 2019) (PMIP3, 2019). ERF data calculated with MIROC4m-AGCM are available from the corresponding author upon request. Computer codes used for the analysis for Figs. 5, 9, and 12 were written in Fortran and they are also available by request except for a random number generator (ran3) taken from Press et al. (1992).

**Author contribution**

This study was developed based on parts of MS's bachelor and master theses at Hokkaido University. MY designed the analysis. MS conducted the initial analysis which was completed by MY. MY prepared the manuscript with contributions from MS. Both authors contributed to the interpretation of the results.

**Competing interests**

The authors declare that they have no conflict of interest.

**Acknowledgements**

We are thankful to Dr. Massonnet and two anonymous reviewers for their useful suggestions which helped us to improve the manuscript substantially. The method for diagnosing the surface temperature change described in Sect. 3b originates from discussions with Dr. Alexandra Laîné in previous works. This study also benefitted from discussions with Prof. Ayako Abe-Ouchi. We acknowledge the World Climate Research Programme's Working Group on Coupled Modelling, which is

responsible for CMIP, and we thank the climate modeling groups (listed in Table 2 of this paper) for producing and making available their model output. For CMIP the U.S. Department of Energy's Program for Climate Model Diagnosis and Intercomparison provides coordinating support and led the development of the software infrastructure in partnership with the

Global Organization for Earth System Science Portals. We thank PMIP for coordinating the experiment and preparing the dataset. We also thank the developers of the freely available software, NCO, CDO, and NCL. The calculation of the radiative forcing with MIROC4m-AGCM was carried out using the JAMSTEC Earth Simulator 3, and the support from the MIROC model development team is appreciated. This study was supported by JSPS KAKENHI Grant Number JP17H06104 and the Arctic Challenge for Sustainability (ArCS) project of MEXT.

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

Table 1 Orbital configurations for the PI and MH experiments. The PI and MH values here represent the values for the years 1850 C.E. and 6000 years before 1950 C.E., respectively, taken from the PMIP3 web page (https://pmip3.lsce.ipsl.fr/). They originate from Berger (1978). Parameters for PI may vary slightly with the model.

| | Eccentricity | Obliquity (°) | Longitude of perihelion from the vernal equinox − 180 (°) |
|---|---|---|---|
| PI | 0.016764 | 23.459 | 100.33 |
| MH | 0.018682 | 24.105 | 0.87 |

Table 2  Models used in the current study and the annual, global and Arctic (north of 60°N) mean surface air temperature changes (°C).

| Model | ΔMH | | ΔRCP4.5 | |
|---|---|---|---|---|
| | Global | Arctic | global | Arctic |
| bcc-csm1-1 | -0.13 | 0.87 | 1.74 | 4.27 |
| CCSM4 | -0.22 | 0.01 | 1.83 | 3.89 |
| CNRM-CM5 | 0.18 | 1.42 | 2.07 | 5.02 |
| CSIRO-Mk3-6-0 | 0.02 | 0.43 | 2.37 | 3.06 |
| FGOALS-g2 | -0.75 | -0.48 | 1.43 | 3.57 |
| FGOALS-s2 | -0.16 | 0.46 | 1.66 | 2.34 |
| GISS-E2-R | -0.10 | 0.77 | 1.34 | 2.45 |
| IPSL-CM5A-LR | -0.13 | 0.25 | 2.37 | 4.84 |
| MIROC-ESM | -0.25 | -0.27 | 2.58 | 6.00 |
| MRI-CGCM3 | -0.02 | 0.81 | 1.70 | 3.84 |
| Mean | -0.16 | 0.43 | 1.91 | 3.93 |

Table 3 A list of the energy flux terms used in Figs. 5 and 12. Row #1 represents the strength of the global mean feedback calculated with local warming sensitivity. Rows #2−10 represent the strength of local feedback calculated with global mean warming sensitivity.

| # | Symbol | Physical meaning |
|---|--------|------------------|
| 1 | S-B | nonlinearity of Stefan-Boltzmann law |
| 2 | alb | surface albedo change |
| 3 | alb*clr_sw | nonlinear effect of surface albedo and clear-sky shortwave radiation changes |
| 4 | clr_sw | clear-sky shortwave radiation change |
| 5 | clr_lw | clear-sky longwave radiation change |
| 6 | cld_sw | shortwave cloud radiative effect |
| 7 | cld_lw | longwave cloud radiative effect |
| 8 | evap | surface latent heat flux via evaporation |
| 9 | sens | surface sensible heat flux |
| 10 | surface | net surface energy flux including latent heat for snow/ice melting and heat exchange with the subsurface |
| 11 | synergy | synergy term for local feedbacks and local warming sensitivity |

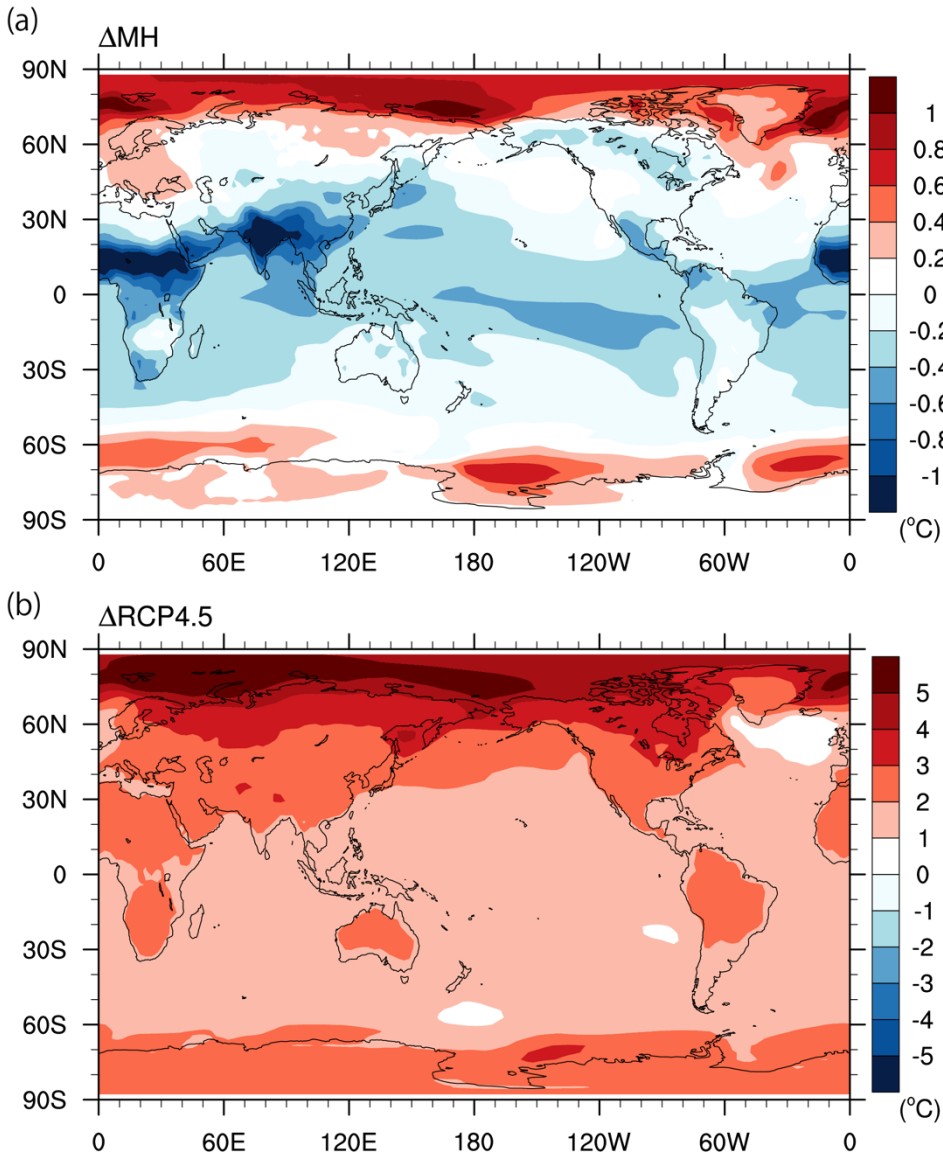

Figure 1  Multi-model mean (all 10 models listed in Table 2) annual mean surface air temperature response (°C): (a) ΔMH; and (b) ΔRCP4.5.

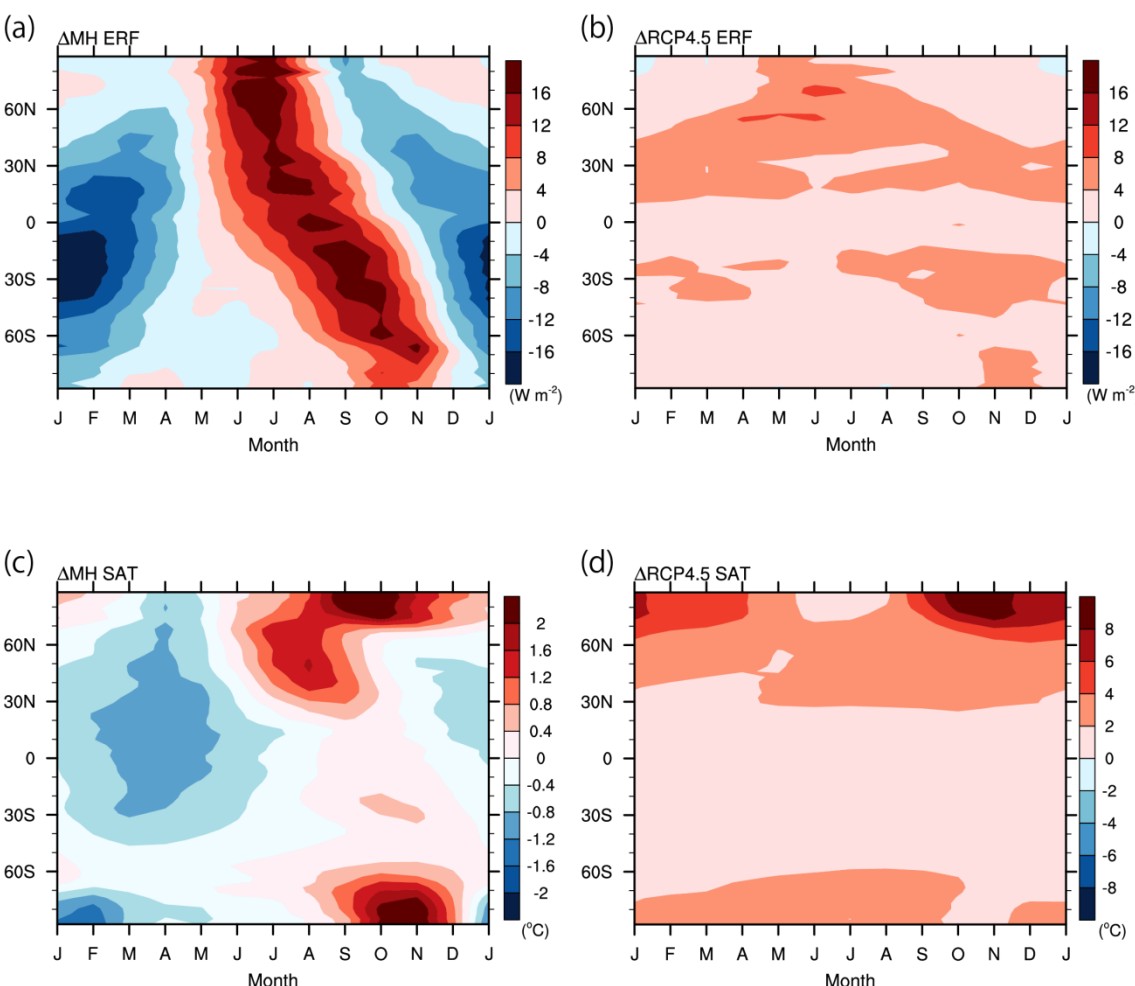

Figure 2   Seasonal progress of the zonal mean effective radiative forcing, ERF (top, W m$^{-2}$) and surface air temperature change (bottom, °C): (a) & (c) ΔMH; and (b) & (d) ΔRCP4.5. The ERF for ΔRCP4.5 is drawn using the data from Yoshimori et al. (2018), and it is computed in the current study for ΔMH. Both ERFs are constructed with a single model, MIROC4m-AGCM (Yoshimori et al., 2018). The surface air temperature changes are the means of all 10 models listed in Table 2.

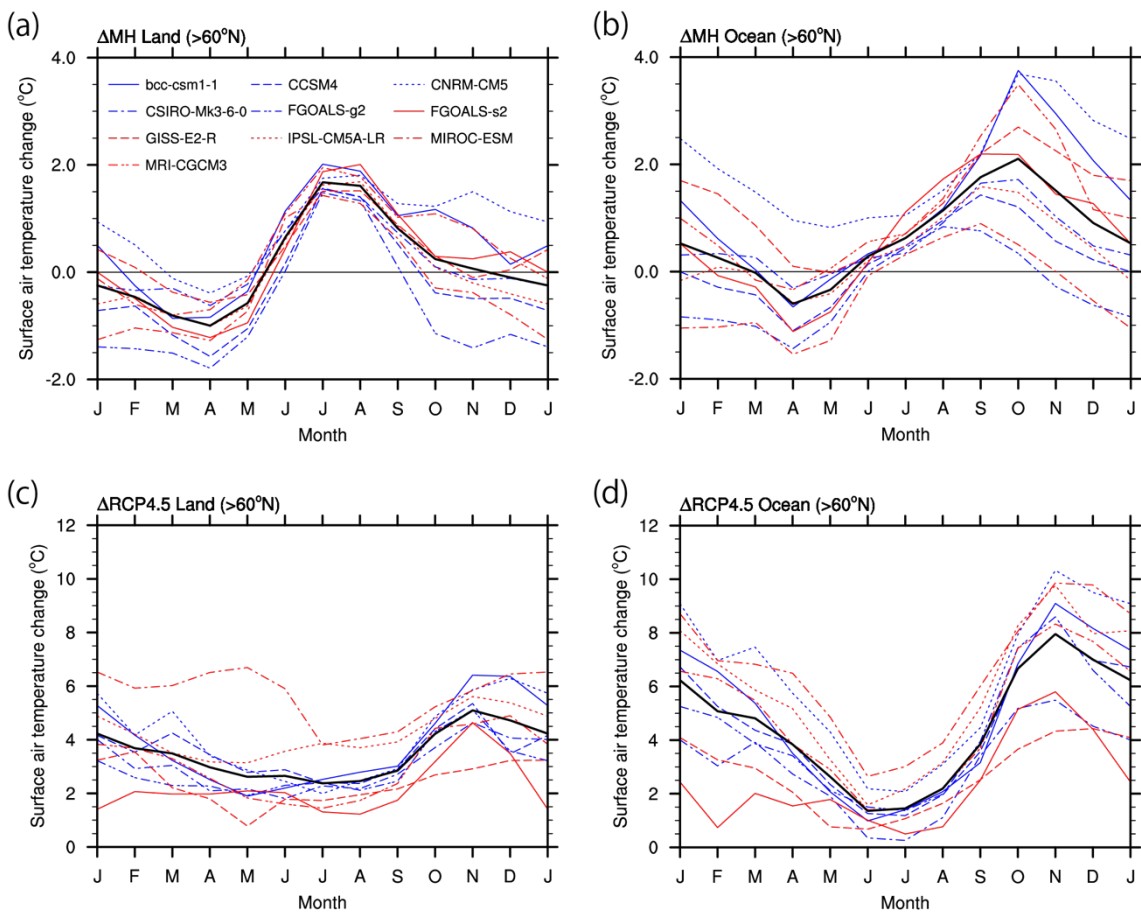

Figure 3  Seasonal progress of the surface air temperature change (°C) in the Arctic (north of 60°N): (a) ΔMH land; (b) ΔMH ocean; (c) ΔRCP4.5 land; and (d) ΔRCP4.5 ocean. Thick black lines show the multi-model mean. Note that the range of vertical axis is different for ΔMH (a and b) and ΔRCP4.5 (c and d).

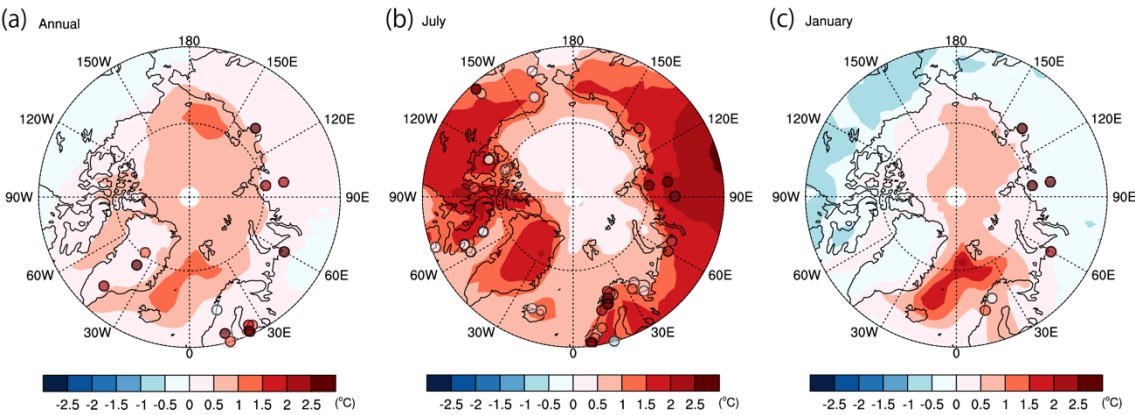

Figure 4  Surface air temperature anomaly (°C) for ΔMH from the simulations (shading) and reconstruction (solid circles): (a) annual mean; (b) July; and (c) January. The reconstruction data are taken from Sundqvist et al. (2010). The mean of all 10 models listed in Table 2 was used.

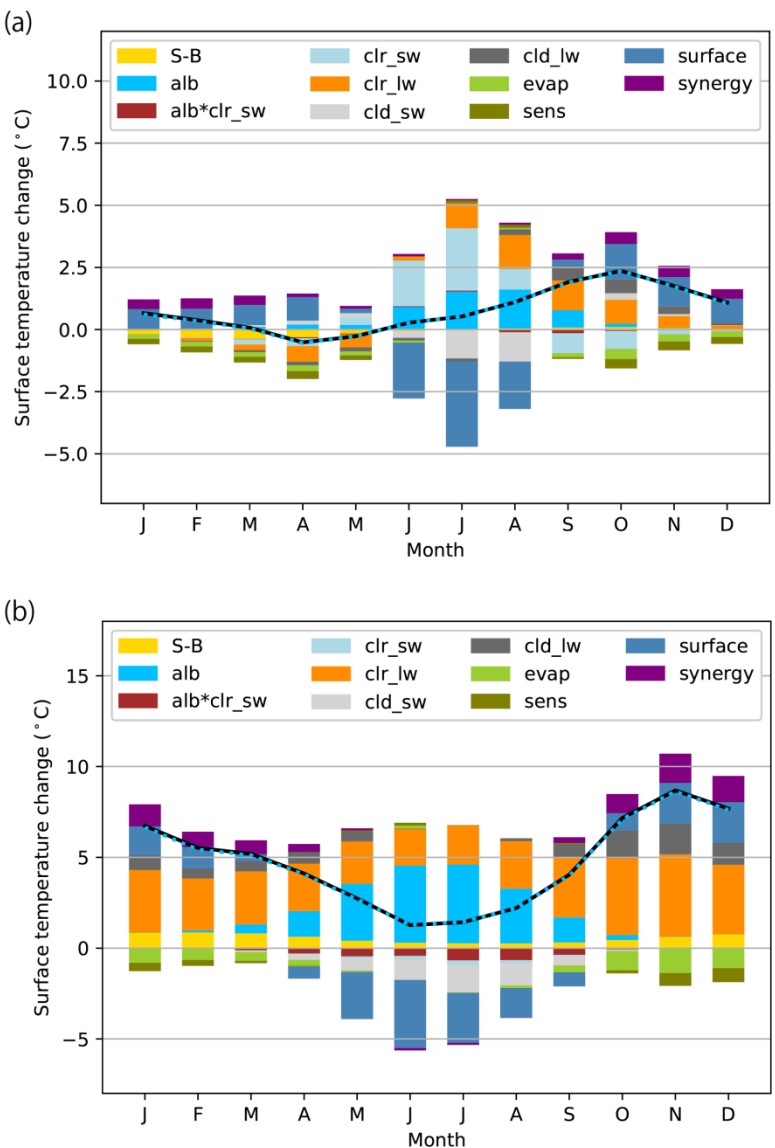

Figure 5 Simulated and diagnosed surface temperature changes (°C) for the ocean (north of 60°N): (a) ΔMH; and (b) ΔRCP4.5. The black polygonal solid lines denote simulated changes and blue polygonal dashed lines denote the sum of the diagnosed partial changes; the two lines are superimposed. The graphs represent the means of all 10 models listed in Table 2. See Table 3 for the interpretation of each component.

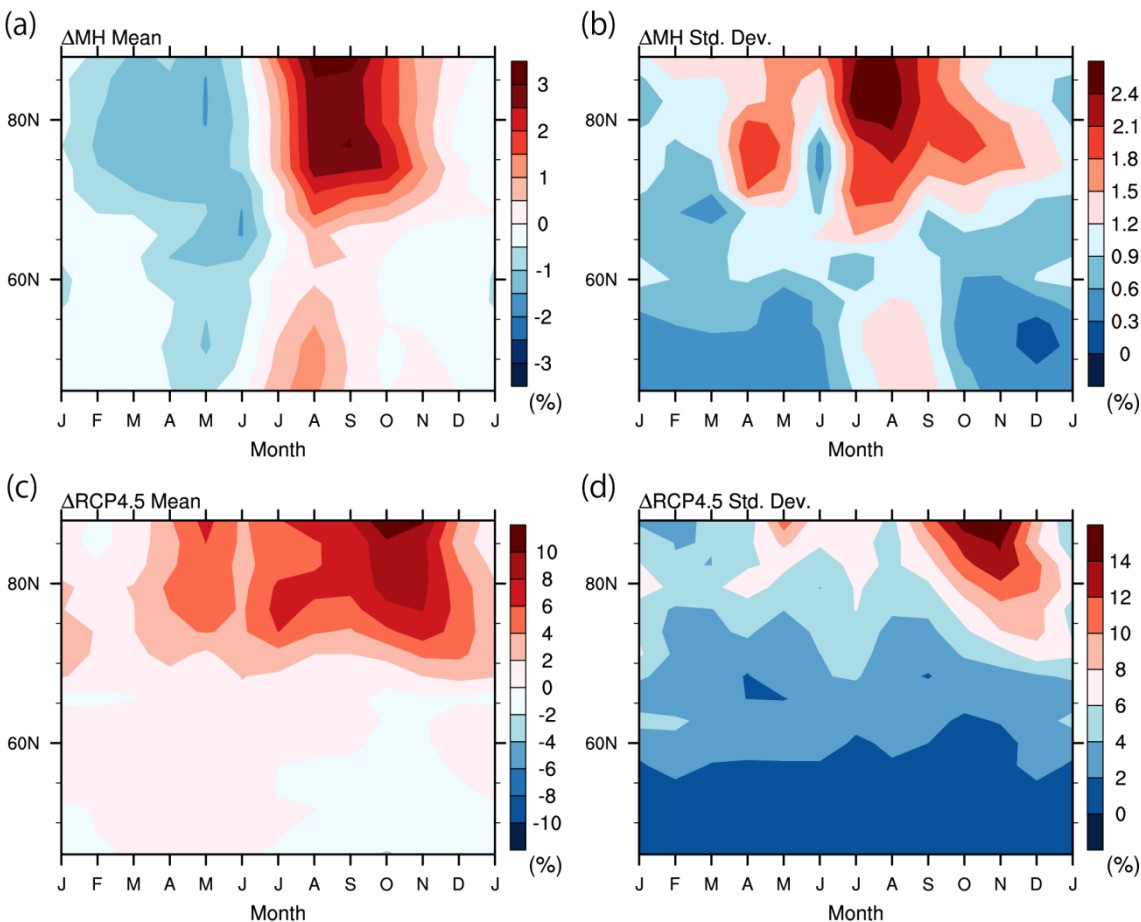

Figure 6  Seasonal progress of the total cloud fraction change (%) over the ocean (north of 60°N): (a) ΔMH ensemble mean; (b) ΔMH ensemble standard deviation; (c) ΔRCP4.5 ensemble mean; and (d) ΔRCP4.5 ensemble standard deviation. All 10 models listed in Table 2 are used.

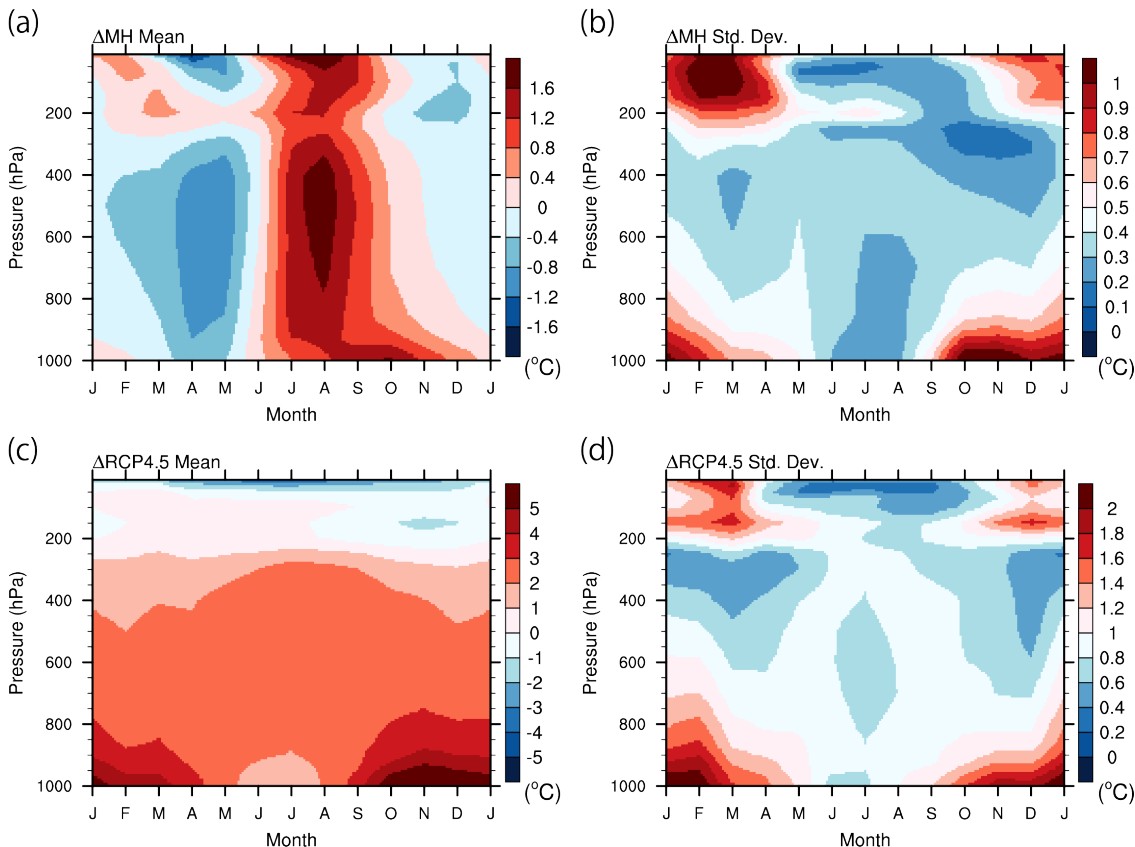

Figure 7  Same as in Fig. 6 but for the air temperature change (°C) (north of 60°N). All 10 models listed in Table 2 are used. (Note that the figure appears blocky compared to Fig. 6 due to the use of a different interpolation scheme in the plotting software which was chosen to avoid a technical issue for pressure coordinate, but it is irrelevant to the data.)

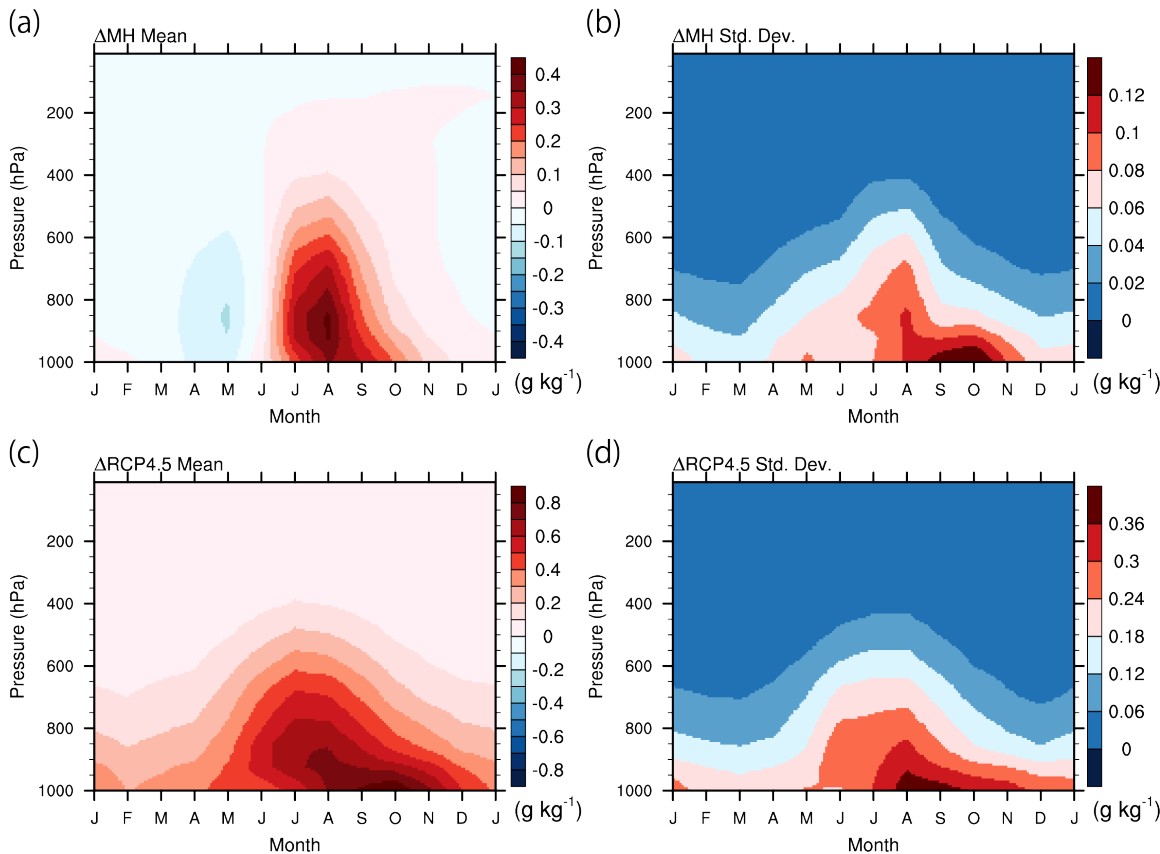

Figure 8  Same as in Fig. 6 but for the specific humidity change (g kg$^{-1}$) (north of 60°N). All 10 models listed in Table 2 are used. (Note that the figure appears blocky compared to Fig. 6 due to the use of a different interpolation scheme in the plotting software which was chosen to avoid a technical issue for pressure coordinate, but it is irrelevant to the data.)

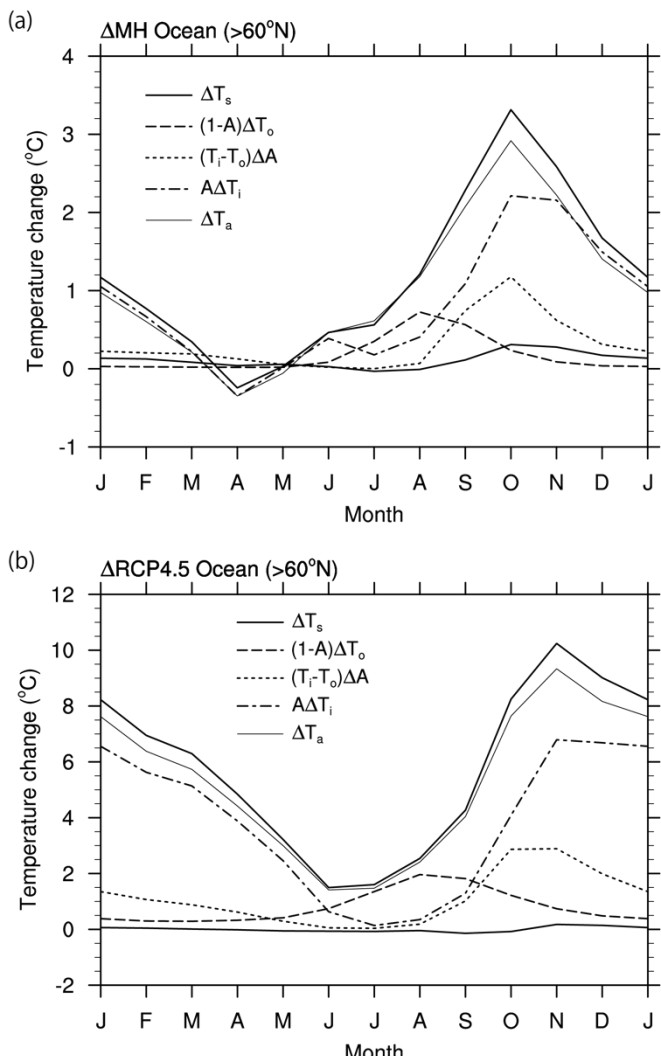

Figure 9 Contribution of the individual components to the surface temperature change (°C) over the ocean (north of 60°N): (a) ΔMH; and (b) ΔRCP4.5. The surface temperature change is decomposed into the components of the SST change $((1 - A)\Delta T_o)$, sea ice concentration change $((T_i - T_o)\Delta A)$, and sea ice surface temperature change $(A\Delta T_i)$. Simulated surface temperature $(\Delta T_s)$ and surface air temperature changes $(\Delta T_a)$ are also plotted for reference. Only 5 models (bcc-csm-1, CCSM4, CNRM-CM5, IPSL-CM5A-LR, and MRI-CGCM3) are used.

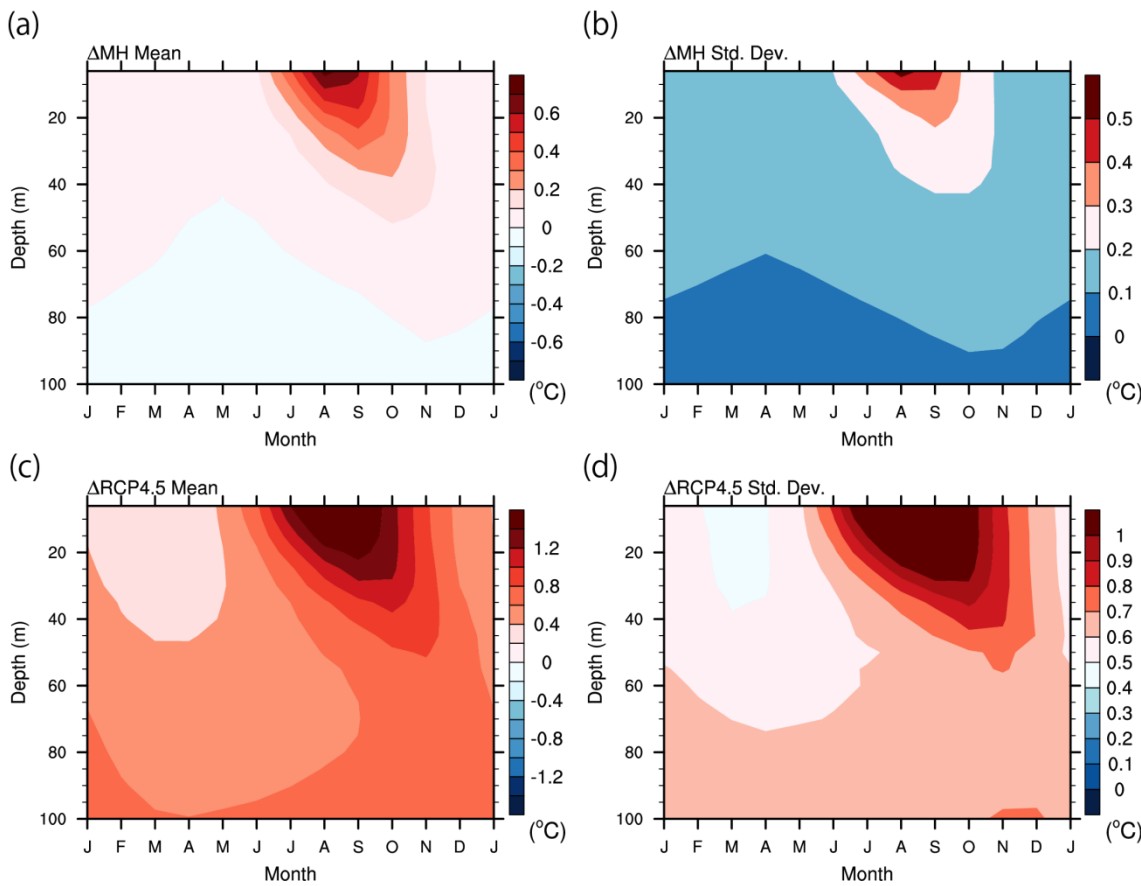

Figure 10 Same as in Fig. 6 but for the upper ocean temperature change (°C) (north of 60°N). 9 models except for FGOALS-g2 listed in Table 2 are used.

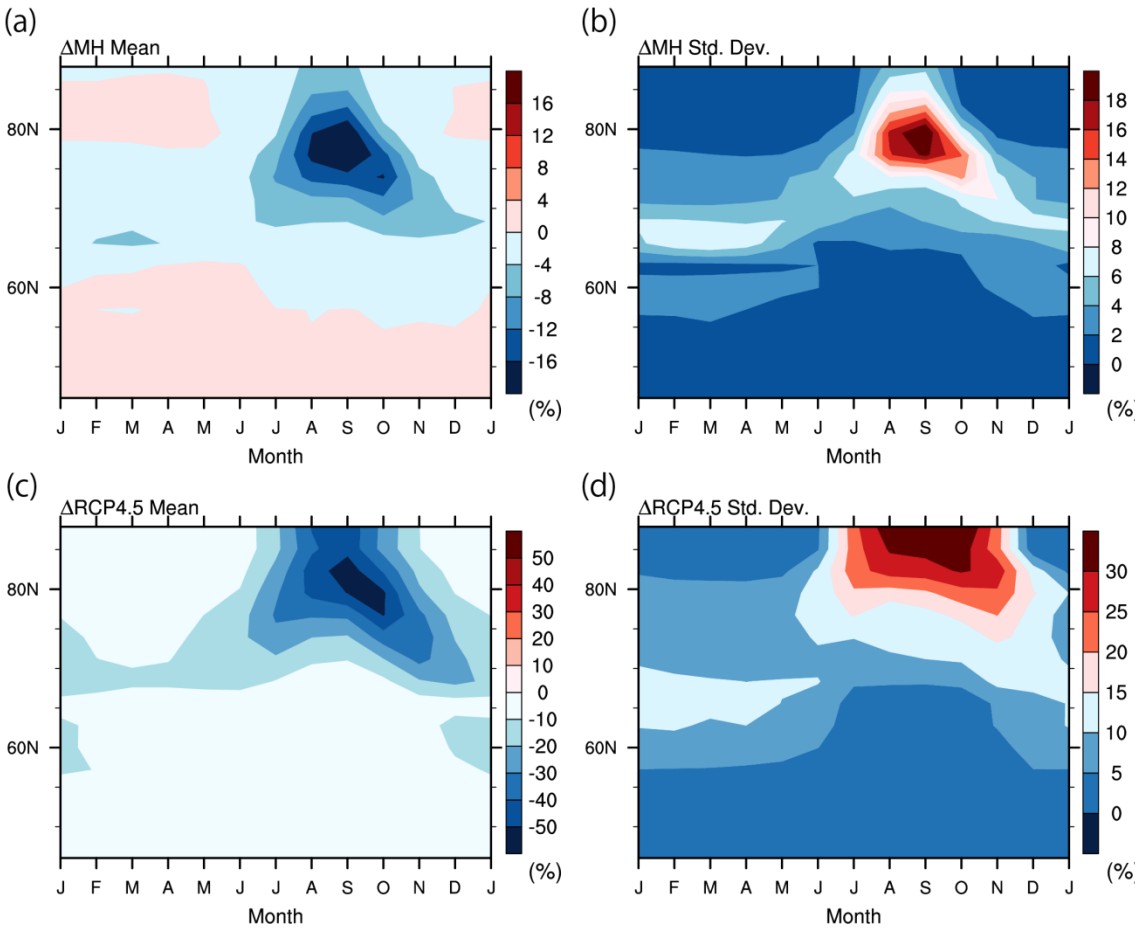

Figure 11  Same as in Fig. 6 but for the sea ice concentration (%) (north of 60°N). All 10 models listed in Table 2 are used.

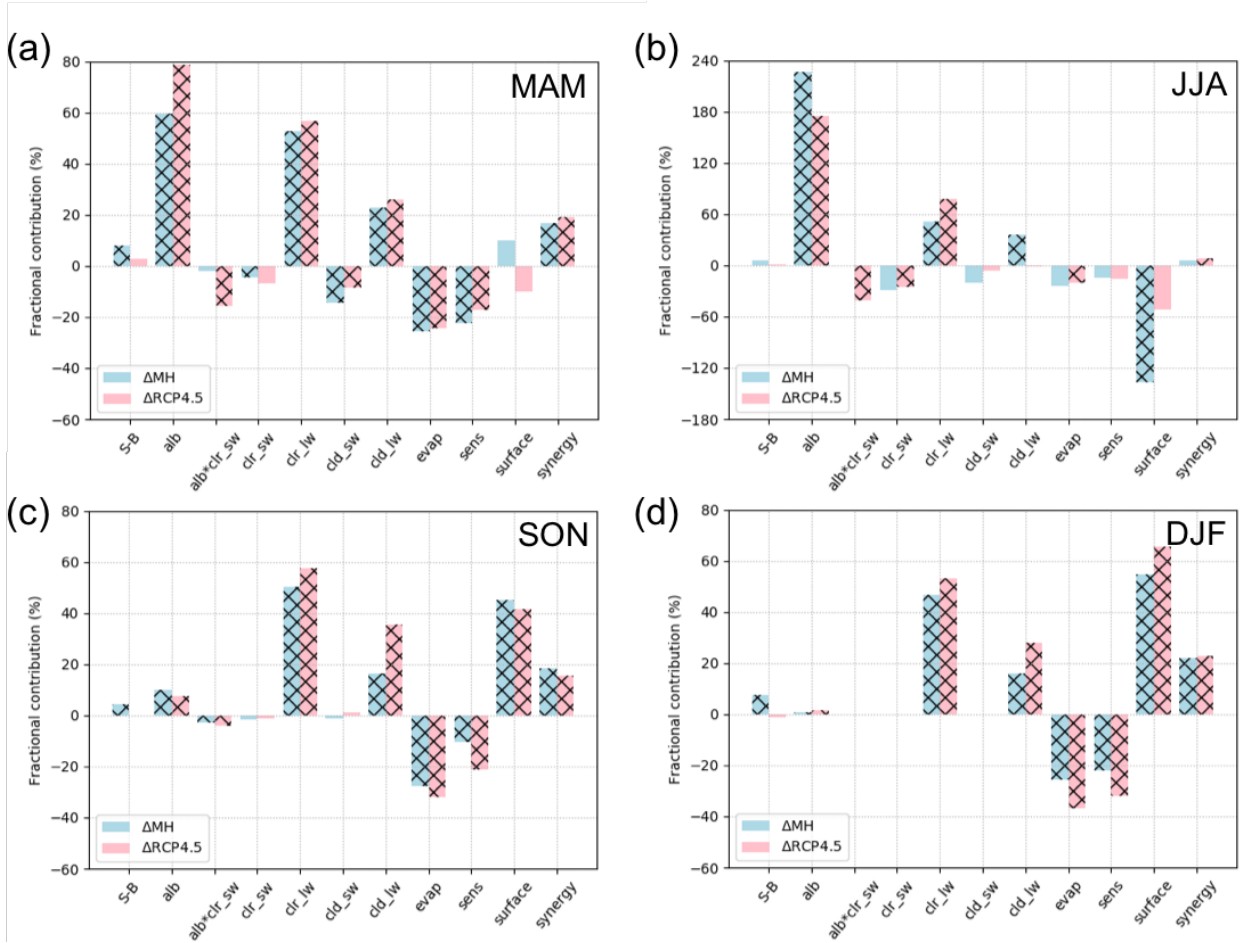

Figure 12 Fractional contribution of individual processes to the model spread in the simulated surface temperature change (%) over the ocean (north of 60°N) for ΔMH and ΔRCP4.5: (a) spring (March-April-May); (b) summer (June-July-August); 
5  (c) autumn (September-October-November); and (d) winter (December-January-February) means. The sum of the bar graphs in the same color for each plot adds up to 100%. The hatching indicates the contribution is statistically significant at the 10% level. All 10 models listed in Table 2 are used. See Table 3 for the interpretation of each component. Note that the vertical scale for (b) is three-fold larger than in the other plots.