# Peer review of "The relevance of mid-Holocene Arctic warming to the future"

_Climate of the Past, 2018_

## Referee Comment (RC1) · Anonymous Referee #1 · 10 Jan 2019

This paper discusses the decomposition of the changes in surface temperature into local and global feedback contributions, related to the different components of the surface energy balance. This decomposition is performed both for Mid-Holocene (compared to pre-industrial) and future warming (under RCP4.5 scenario).

As a general comment I would say that I found the paper difficult to read, although I cannot figure out exactly the reason (either the topic or the language).

I also found that the discussion is not really a discussion, but rather a perspective and conclusion. After reading the manuscript, and although I acknowledge similarities in climate changes between MH and the future, I still do not understand the 'relevance of mid-Holocene Arctic warming to the future '. This should be the major item in the discussion.

[Figure]

The conclusion of the paper is not very new. It has already been repeated many times that 'improvement of the ability of the model to simulate the pas twill increase the confidence in their ability to simulate the future'. I would suggest to identify a 'crispier' conclusion.

The manuscript is missing a data availability section. Moreover, data citations are also missing (in addition to the references that are indeed given). This is the case, at least for the data of Bartlein et al (2011) and Sundqvist et al (2010). Moreover, the code used to extract the values displayed in figure 5 should be made publicly available as well (with a reference in the data availability section).

Specific comments.

P1-l29 : is it solar forcing?

P2-l1 : I assume that the 'scenario' refers to RCP scenarios. This should be made clear.

P4-l16 and P7-l13 : there is a reference to Sect. 3a, which does not exist (at least as such).

P6-l33 : According to my reading of the figure, the simulated warming only occurs in the northern North Atlantic and Arctic oceans, where there is no data. It is therefore very difficult to say if it is under- or over-estimated. Or do the authors call 'warming' the negative values in the figure?

P7-l16 : 'plays an important role'. According to my reading, this is only true in JJA.

P7-l32 : 'exhibits a large contribution'. This does not really seem to be the case for MH.

P8-l9 : could the authors make the label coherent (Dtas in the text, Dta in the figure).

P10-l11 : PMIP3 instead of PMIM3

[Figure]

P10-l22 : The authors should make their conclusion readable by itself. It should be said that the Arctic warming is for the future (under RCP4.5).

P10-l33 : 'seeking possible analogues between physical processes in the past and future climate'. Do the authors mean that the climate processes are time dependent? I thought that they were based on basic physical principles valid through time. Moreover, as we do not know the future climate it is hard to look for analogues there and then.

P15 : A reference is missing here.

P 21 : The figure is misleading because the Y-axis (scale) is not the same for MH and RCP4.5.

P23-l4 : I do not see two (black and blue dashed) lines. Are they exactly superimposed? In that case, this should be mentioned in the caption.

P24-26 : Figures 6-8 are not using the same number of models. (1) the name of the models used should be mentioned. (2) Not using all the models (and not always the same models) may introduce a bias in the interpretation. Would the conclusion be the same if only the models (and their outputs) available for all the figures were used?

---

## Author Comment (AC1) · 15 Jan 2019

Thank you very much for carefully reading the manuscript and for bringing up some important points. Perspective on the revision is provided below.

As to the discussion, we will revise the manuscript and place more emphasis on why we consider the knowledge of mid-Holocene Arctic warming is useful when future Arctic warming projections are made. What we consider the most important point is that the two different time periods (MH and future) are relevant through similarities of dominant processes in shaping the Arctic warming even though the radiative forcing and climate response are not alike. It is an indirect relevance, but this view has not been explored much previously, particularly for the Arctic warming. It is not a priori trivial whether the

[Figure]

dominant processes are common in response to two different climate forcers among many processes, and we show a link in mechanism between the two climates.

As to the conclusion, we will revise the manuscript so that it highlights more on what is new here. As the reviewer pointed out, the way it was rewritten may be too generalized with less emphasis on specific findings here. As the reviewer says, the phrase "improvement of the ability of the model to simulate the past will increase the confidence in their ability to simulate the future" may have been used repeatedly, but it is often used in an intuitive context, rather than with rigorous analysis, particularly for the Arctic warming. What we consider new findings here are 1) the similarities in processes responsible for the Arctic warming between MH and future despite of very different radiative forcing pattern; and 2) those processes are also contribute to the model spread in the Arctic warming between the two periods. Mechanism of MH Arctic warming and future in multi-models have not been investigated in parallel previously and to the extent conducted here. In addition, the similarities in processes are not shown previously. Furthermore, the warming processes are detailed by separating the sea surface temperature and sea ice surface temperature changes in Fig. 7 which adds clear physical understanding to the conventional surface energy balance feedback analysis. We will make the conclusion sharper.

We will add Data Availability section. We will also make the main part of the program used in the analysis for Fig. 5 publicly available.

Specific comments.

>P1-l29 : is it solar forcing?

Marshall et al. (2014) suggests stratospheric ozone forcing. To be specific, we will change it to "stratospheric ozone change and cloud feedback play roles".

>P2-l1 : I assume that the 'scenario' refers to RCP scenarios. This should be made clear.

Change will be made to "RCP scenario".

>P4-l16 and P7-l13 : there is a reference to Sect. 3a, which does not exist (at least as such).

"3a" should be "3.1". Thank you.

>P6-l33 : According to my reading of the figure, the simulated warming only occurs in the northern North Atlantic and Arctic oceans, where there is no data. It is therefore very difficult to say if it is under- or over-estimated. Or do the authors call 'warming' the negative values in the figure?

We will change it to "the warming indicated by the reconstruction is not captured by the model mean in January as well as in the annual mean."

>P7-l16 : 'plays an important role'. According to my reading, this is only true in JJA.

We will change it to "plays an important role in June-July-August (and in September-October in opposite sign)".

>P7-l32 : 'exhibits a large contribution'. This does not really seem to be the case for MH.

The reviewer is correct. We will clarify this point.

>P8-l9 : could the authors make the label coherent (Dtas in the text, Dta in the figure).

The correction will be made to the text.

>P10-l11 : PMIP3 instead of PMIM3

Will be corrected. Thank you.

>P10-l22 : The authors should make their conclusion readable by itself. It should be said that the Arctic warming is for the future (under RCP4.5).

We will revise the conclusion taking the reviewer's comment into account.

>P10-l33 : 'seeking possible analogues between physical processes in the past and future climate'. Do the authors mean that the climate processes are time dependent? I thought that they were based on basic physical principles valid through time. Moreover, as we do not know the future climate it is hard to look for analogues there and then.

While physical principles are same throughout the time, what we meant was that it is not trivial that the dominating processes for the climate variations are the same for different climate forcing and change cases. We will rephrase the sentence.

>P15 : A reference is missing here.

We will add the reference.

>P 21 : The figure is misleading because the Y-axis (scale) is not the same for MH and RCP4.5.

We will add the note on the caption. The difference in magnitude does not preclude the use of these two different time periods, and rather it is of interest that such different climate responses still share the similar dominant processes.

>P23-l4 : I do not see two (black and blue dashed) lines. Are they exactly superimposed? In that case, this should be mentioned in the caption.

They are black polygonal solid-line and blue polygonal dashed-line. We will make the caption more precise.

>P24-26 : Figures 6-8 are not using the same number of models. (1) the name of the models used should be mentioned. (2) Not using all the models (and not always the same models) may introduce a bias in the interpretation. Would the conclusion be the same if only the models (and their outputs) available for all the figures were used?

(1) All model names were given in Table 2, but the models used for Fig. 7 was only written in text. We will refer Table 2 for Figs. 6 and 8, and write names explicitly for Fig. 7 in the caption. (2) The main results shown in Figs. 5 and 10 are benefitted

the most by using the models as many as possible (10 models), but all variables are available only for 5 models: 5 models are missing for Fig. 7 and 1 model is missing for Fig. 8 (It was mistakenly written that 2 models were missing in the caption of Fig. 8. We will correct it). We checked the consistency of Figs. 5, 7 and 8 by reducing the model numbers to 5. Figs. 5 and 8 were not qualitatively affected by this reduction. We also checked all figures by reducing the model numbers to 5: a few terms lost their statistical significance in Fig. 10, but the conclusion remains the same.

---

## Referee Comment (RC2) · Massonnet (Referee) · 31 Jan 2019

Review of

The relevance of mid-Holocene Arctic warming to the future

by M. Yoshimori and M. Suzuki.

In this study, the authors conduct a diagnostic surface balance analysis based on output from the PMIP3 and CMIP5 simulations. They wish to test the extent to which past Arctic warming could be used as an analogue for future warming, which could then make the basis for a more objective model selection. The authors find that despite different forcing mechanisms, several common feedbacks operate between the two periods, making the case that these periods can indeed be compared to one another.

The paper is interesting to read but is quite descriptive: the differences between the MH-PI and RCP4.5-Historical simulations are stated and described, but not a lot of attention is given to try to explain why patterns differ and, more importantly, why this would have implications for the scientific community. The conclusions fall a bit short, for example. The authors explain that the MH period could be used to evaluate the models, but they do not state what type of constraints could be applied. Based on their results, can the authors make a step forward and come with recommendations on such constraints?

A few general comments:

* The analysis relies on four types of runs: Mid-Holocene (MH), Pre-industrial (PI), Historical and RCP4.5. I understand that MH simulations are taken from PMIP3, and so are PI simulations. I understand that Historical and RCP4.5 simulations are taken from CMIP5. Is that correct? Something confusing is that the authors write that "For the MH and PI simulations, we use monthly climatological data averaged over periods longer than a century, which were archived as part of the CMIP5 dataset" but also write that MH simulations were taken from PMIP3: "The MH simulation was designed and coordinated by the PMIP3 project". Could the authors clarify this at p. 2, line 30 (I did not find the explanations very clear).

* There is an important negative feedback that is not mentioned in the study: the negative ice growth-ice thickness feedback, which states that sea ice grows faster when it is thin. The existence of this feedback is a safeguard for sea ice, which would otherwise disappear much faster due to the positive albedo feedback. I'm unclear if the aforementioned negative feedback is covered at all by the authors and if so, to which term of Eq. 2 it belongs.

* There are a two references missing that deal with high-latitude changes and the role of feedbacks, that I think should appear in the text: - DOI:10.1038/s41598-017-04623-7 - DOI: 10.1038/s41467-018-04173-0

Specific comments (Syntax: 22-03 = line 22, page 3)

19-01: "indirect atmospheric stratification" might be unclear to many. Please rephrase or explain. 06-02: "time periods" –> "periods" (a period is always referring to time) 07-02: "discouraged general comparisons": do you mean that the studies found that comparisons were not simple to make? Please rephrase. 23-02: "time periods" –> cf. 06-02. 24-02: The last sentence of the paragraph is not quite clear; consider removing it. 26-03: "effect" –> effects 22-04: "ts" shoud be T_s in mathematical form. 07-05: Why using the \Lambda sign for temperature differences, and not \Delta T? It is not clear how \Lambda relates to Eqs (6) and (4). 28-05: Can you elaborate on how the ERF was computed precisely? It is said that an AGCM was used, but which one? What was the exact setup? It would be impossible to reproduce your results if the readers do not have this information.

---

## Author Comment (AC2) · 8 Feb 2019

Thank you very much for carefully reading the manuscript and for pointing out some of the messages that need to be sharpened. Perspective on the revision is provided below.

As to the differences between MH and future, and to the implication for the scientific community, we will add some discussions and crystalize the message. As the main goal of this paper is to show the link between MH and future Arctic warming, there are less focus on the differences as they are naturally expected from the different radiative forcing patterns. What was not pointed out in previous studies, to the extent analyzed here for multi models, is the transfer of extra energy absorbed in the ocean during
summer to the heat release from the ocean during winter, and consequent amplified warming occur in a similar way in response to different types of radiative forcing. This notion is also valuable to understand the Arctic response in much wider paleoclimate conditions. On the other hand, the reviewer's point is valid in that the difference is less emphasized: early spring response is particularly distinct between MH and future forcing cases. This needs to be mentioned even though the current multi-model analysis does not identify the exact mechanism. As in the response to the reviewer 1, we will sharpen the conclusion so that new findings and implication become clearer. We do not claim any new 'emergent constraints' in the current study although that would offer more practical implication. We believe that the application of such constraint should go hand in hand with mechanism understanding, statistical identification of the link between the past and the future (e.g., Schmidt et al., 2014), and paleoclimate proxy searches suitable to constrain the link. In our view, the community is not ready to apply such an integrated approach using the MH Arctic state with confidence. Nevertheless, our study suggests that proxy records quantitatively measuring winter Arctic warming in MH (relative to the preindustrial) would have a potential as a constraint, based on our mechanism understanding of how winter warming is amplified commonly between MH and future. We hope the current study provides a step towards such an ultimate goal for the community.

Reply to general comments:

1. We apologize for the confusion between PMIP3 and CMIP5. The MH experiment was designed by PMIP3, and that was endorsed as a part of CMIP5. All the data were downloaded from CMIP5 data base. We will clarify this point.

2. The negative ice growth-ice thickness feedback is not quantified explicitly in the current analysis. Therefore, it does not appear in the decomposed terms in Equation (2) although they are closely linked to the sea ice related terms including the magnitude of albedo feedback (a function of ice cover among others) and heat release from the ocean (a function of ice thickness among others). Our analysis is based on the

surface energy balance as in many other previous studies. The quantification of ice thickness feedback would require energy budget analysis for sea ice itself and probably for mixed-layer of the ocean as well. This does not mean that we think the feedback is unimportant. We will mention this point.

3. Thank you for pointing out uncited references. We will cite them in the revised manuscript.

Reply to specific comments:

>19-01: "indirect atmospheric stratification" might be unclear to many. Please rephrase or explain.

>06-02: "time periods" –> "periods" (a period is always referring to time)

>07-02: "discouraged general comparisons": do you mean that the studies found that comparisons were not simple to make? Please rephrase.

>23-02: "time periods" –> cf. 06-02.

>26-03: "effect" –> effects

We will rephrase/change these expressions as suggested.

>24-02: The last sentence of the paragraph is not quite clear; consider removing it.

We will remove it.

>22-04: "ts" shoud be T_s in mathematical form.

We will correct this.

>07-05: Why using the \Lambda sign for temperature differences, and not \Delta T? It is not clear how \Lambda relates to Eqs (6) and (4).

We will replace \Lambda by \Delta T.

>28-05: Can you elaborate on how the ERF was computed precisely? It is said that an

AGCM was used, but which one? What was the exact setup? It would be impossible to reproduce your results if the readers do not have this information.

The model information was only given in the figure caption. We will move this into the text, and also add more precise description as to the setting of the ERF computation.

---

## Referee Comment (RC3) · Anonymous Referee #3 · 18 Feb 2019

General comment

This manuscript proposes an in depth analysis of the different radiative and turbulent latent and sensible heat fluxes terms that constraint the seasonal changes in surface temperature in the Arctic. The analysis considers the mid-Holocene climate and the RCP4.5°C scenario for the future, with the objective to derive emerging constraints from the mid-Holocene climate that can be used to assess the results of future climate projections. This analysis is interesting, but the conclusion is not strong enough about the analogies between the two periods and what can be done out of it. It is only during the ice melting period, when albedo decreases and water vapor increases in the atmosphere, that similar feedbacks occur. The forcing factors are very different between the two periods. Even though the different elements are found in the text, similarities and

differences could be better discussed. The abstract could also be more informative on the results and better stress the role of the clear sky long wave radiation. The different figures are difficult to follow, because there is no direct relationship between the names of the different terms plotted in figure 5 (a key figure in this manuscript) and the decomposition done using equation 2 to 7. I therefore consider that this manuscript is worth publishing, but that an effort should be made to clarify the expression of the different terms and better explain their role. The discussion should also be enlarged, so that the paper more clearly address the point listed in the title.

Other comments: - P2 make sure you properly refer CMIP or PMIP everywhere. - P 3 l 15. And section 4.3. The comparison of the MH results with observations is not fully used in the manuscript. Is there a way to go one step further by provided an evaluation that could really inform on the relevant processes between past and future? - P3bl 29. Could you provide an order of magnitude of the uncertainty related to emissivity for models that have a variable emissivity? - P 4 2 Is the equation correct for S? - P4 l 15 what do you call sect. 3a? - P4 end of section 3.1. It could be worth mentioning that the approach is direct because there is no change in land-sea mask between the different simulations. - P4. L 27 are your referring to ice concentration or to ice fraction? - P 5 Would it be possible to rewrite equation 7 so that there is a more direct link with temperature ? or use one example to fully explain what is done and the strength of the diagnosis. This could also be needed to present the different terms of equation (4) and make sure there is no ambiguity on global or local anomaly (or their relative strength). - P4 l 8. May be you could site Hewitt and Mitchell 1997 for the definition of the MH insolation forcing. - P7 There is a large emphasize on clouds before showing the effect of lw_clr. This later term reflects both changes in water vapor and in atmospheric lapse rate. The cloud cld_effect arrives later (in season) compared to albedo and lw_clr. I would suggest reconsidering the way the whole section is written, to better discuss the relationship between the different terms and their monthly evolution. - P8 section 4.5. I am not entirely convinced that OND are the best months to look at to infer model spread. Sea-ice and temperature result certainly of what happens during the preceding

months in terms of forcing and feedbacks. This needs to be clarified. - P9 section 4.5 I am lost in the call to the different figures. Figure 10 also show a large model spread in the lw_clr, not only in clouds. This should be highlighted. The cloud cover is important but results certainly from the other conditions: sea-ice fraction, temperature, lapse rate, water vapor, changes in atmospheric convection or large scale condensation. This should be discussed, at least to tell when there is an analogy or not between the different feedbacks between mid-Holocene and future climates.

---

## Author Comment (AC3) · 14 Mar 2019

Thank you very much for carefully reading the manuscript and for bringing up some important points. In the following, reviewer's comments are indicated by [RC]. Response to the comment and perspective on the revision are indicated by [AC].

[RC] This paper discusses the decomposition of the changes in surface temperature into local and global feedback contributions, related to the different components of the surface energy balance. This decomposition is performed both for Mid-Holocene (compared to pre-industrial) and future warming (under RCP4.5 scenario). As a general comment I would say that I found the paper difficult to read, although I cannot figure out exactly the reason (either the topic or the language).

[Figure]

[AC] Taking also the suggestions by other reviewers into account, we will change following points to improve the readability.

(1) We will change the abstract to be more informative with emphasis on the specific new findings.

(2) We will write terms in Eq. (4) explicitly after combining with Eq. (2), so that each term corresponds exactly to the description in Table 3 and each component in Figs. 5 and 10.

(3) We replace Lambda in Eq. (7) by T so that it is obvious that the symbol represents temperature.

(4) In Sect. 4.3 in "Results", we will describe the results season by season first, and then state important points afterward so that the reader can grasp the overall results in the sequential order first.

(5) In Sect. 4.5 in "Results", we will describe the results season by season first, and then state important points afterward for the same reason as (4).

[RC] I also found that the discussion is not really a discussion, but rather a perspective and conclusion. After reading the manuscript, and although I acknowledge similarities in climate changes between MH and the future, I still do not understand the 'relevance of mid-Holocene Arctic warming to the future '. This should be the major item in the discussion.

[AC] We will enlarge the discussion and conclusion with emphasis on the relevance in the Arctic response between the MH and future (RCP4.5). The discussion will be substantially enlarged with separate points (1) in terms of the ensemble mean response, and (2) in terms of the model spread. We will also increase discussion for the difference between the MH and future (when and how).

[RC] The conclusion of the paper is not very new. It has already been repeated many times that 'improvement of the ability of the model to simulate the past will increase the

confidence in their ability to simulate the future'. I would suggest to identify a 'crispier' conclusion.

[AC] We will make the conclusion more specific and reduce weight for general statements. The main points will be:

(1) It is found that many of the dominant processes that amplify Arctic warming over the ocean from late autumn to early winter are common between the two periods, despite the difference in the source of the forcing (insolation vs. greenhouse gases).

(2) A chain of processes responsible for the warming trend from summer to autumn is elucidated by the decomposition to factors associated with sea surface temperature, ice concentration, and ice surface temperature changes.

(3) The downward clear-sky longwave radiation is one of major contributors to the model spread throughout the year. Other controlling terms vary with the season, but they are similar between the MH and the future in each season.

(4) The MH Arctic change may not be directly relevant to the future in some seasons (spring in particular) when the temperature response differs, but it is still useful to constrain the future Arctic projection (partly new addition to the original manuscript).

(5) The significant cross-model correlation found between summer albedo feedback and autumn-winter surface temperature response in both forcing cases suggests that feedbacks in preceding seasons, sea ice cover in particular, should not be overlooked as a constraint (new addition to the original manuscript).

[RC] The manuscript is missing a data availability section. Moreover, data citations are also missing (in addition to the references that are indeed given). This is the case, at least for the data of Bartlein et al (2011) and Sundqvist et al (2010). Moreover, the code used to extract the values displayed in figure 5 should be made publicly available as well (with a reference in the data availability section).

[AC] We will add Data Availability section. We will also make the computer codes used

for the analysis in Fig. 5 available upon acceptance of the paper and upon request.

[RC] Specific comments. P1-l29 : is it solar forcing?

[AC] Marshall et al. (2014) suggests stratospheric ozone forcing. To be precise, we will change to "stratospheric ozone change and cloud feedbacks play some roles".

[RC] P2-l1 : I assume that the 'scenario' refers to RCP scenarios. This should be made clear.

[AC] Change will be made to "RCP scenario".

[RC] P4-l16 and P7-l13 : there is a reference to Sect. 3a, which does not exist (at least as such).

[AC] "3a" should be "3.1", and will be corrected.

[RC] P6-l33 : According to my reading of the figure, the simulated warming only occurs in the northern North Atlantic and Arctic oceans, where there is no data. It is therefore very difficult to say if it is under- or over-estimated. Or do the authors call 'warming' the negative values in the figure?

[AC] We will change it to "the warming indicated by the reconstruction is not captured by the model mean in January as well as in the annual mean."

[RC] P7-l16 : 'plays an important role'. According to my reading, this is only true in JJA.

[AC] We will make it more precise.

[RC] P7-l32 : 'exhibits a large contribution'. This does not really seem to be the case for MH.

[AC] The reviewer is correct. We will clarify this point.

[RC] P8-l9 : could the authors make the label coherent (Dtas in the text, Dta in the figure).

[AC] The correction will be made to the text.

[RC] P10-l11 : PMIP3 instead of PMIM3

[AC] Will be corrected. Thank you.

[RC] P10-l22 : The authors should make their conclusion readable by itself. It should be said that the Arctic warming is for the future (under RCP4.5).

[AC] We will make the conclusion readable by itself by adding some words.

[RC] P10-l33 : 'seeking possible analogues between physical processes in the past and future climate'. Do the authors mean that the climate processes are time dependent? I thought that they were based on basic physical principles valid through time. Moreover, as we do not know the future climate it is hard to look for analogues there and then.

[AC] While physical principles are same throughout the time, what we meant is, it is not trivial that the dominating processes for the climate variations are the same for different climate forcing and change cases. We will rephrase the sentence.

[RC] P15 : A reference is missing here.

[AC] We will add the reference.

[RC] P 21 : The figure is misleading because the Y-axis (scale) is not the same for MH and RCP4.5.

[AC] We will add the note on the caption. The difference in magnitude does not preclude the use of these two different time periods, and rather it is of interest that such different climate responses still share the similar dominant processes.

[RC] P23-l4 : I do not see two (black and blue dashed) lines. Are they exactly superimposed? In that case, this should be mentioned in the caption.

[AC] They are black polygonal solid-line and blue polygonal dashed-line. We will make the caption more precise (and text).

[RC] P24-26 : Figures 6-8 are not using the same number of models. (1) the name of the models used should be mentioned. (2) Not using all the models (and not always the same models) may introduce a bias in the interpretation. Would the conclusion be the same if only the models (and their outputs) available for all the figures were used?

[AC]

(1) All model names were given in Table 2, but the models used for Fig. 7 was only written in text. We will refer Table 2 for Figs. 6 and 8, and write names explicitly for Fig. 7 in the caption.

(2) The main results shown in Figs. 5 and 10 are benefitted the most by using the models as many as possible (10 models), but all variables are available only for 5 models: 5 models are missing for Fig. 7 and 1 model is missing for Fig. 8 (It was mistakenly written that 2 models were missing in the caption of Fig. 8. We will correct it). We checked the consistency of Figs. 5, 7 and 8 by reducing the model numbers to 5. Figs. 5 and 8 were not qualitatively affected by this reduction. We also checked all figures by reducing the model numbers to 5: a few small terms lost their statistical significance in Fig. 10, but the conclusion remains the same.

---

## Author Comment (AC4) · 14 Mar 2019

Thank you very much for carefully reading the manuscript and for pointing out some of the messages that need to be sharpened. In the following, reviewer's comments are indicated by [RC]. Response to the comment and perspective on the revision are indicated by [AC].

[RC] In this study, the authors conduct a diagnostic surface balance analysis based on output from the PMIP3 and CMIP5 simulations. They wish to test the extent to which past Arctic warming could be used as an analogue for future warming, which could then make the basis for a more objective model selection. The authors find that despite different forcing mechanisms, several common feedbacks operate between the two

periods, making the case that these periods can indeed be compared to one another.

The paper is interesting to read but is quite descriptive: the differences between the MH-PI and RCP4.5-Historical simulations are stated and described, but not a lot of attention is given to try to explain why patterns differ and, more importantly, why this would have implications for the scientific community.

[AC] In the original manuscript, the similarity in feedbacks in particular season (autumn) might have been too emphasized, and less attention was paid to the difference between the MH and future. We will enlarge the discussion and conclusion with emphasis on the relevance in the Arctic response between the MH and future (RCP4.5). The discussion will be substantially enlarged with separate points (1) in terms of the ensemble mean response, and (2) in terms of the model spread. We will also discuss not only the similarities but also for the difference between the MH and future (when and how). A particular attention will be paid to spring when the ensemble mean response differs between the two periods.

[RC] The conclusions fall a bit short, for example. The authors explain that the MH period could be used to evaluate the models, but they do not state what type of constraints could be applied. Based on their results, can the authors make a step forward and come with recommendations on such constraints?

[AC] We do not claim any new 'emergent constraints' in the current study although that would offer more practical implication. We believe that the application of such constraint should go hand in hand with mechanism understanding, statistical identification of the link between the past and the future (e.g., Schmidt et al., 2014), and paleoclimate proxy searches suitable to constrain the link. Nevertheless, we will add "recommendations" that the seasonal evolution of surface temperature response (cold season in practice) and likely summer sea ice cover are likely useful constraints based on the current analysis. In addition, we will make the conclusion (and abstract) more specific so that the messages become clearer. The main points will be:

(1) It is found that many of the dominant processes that amplify Arctic warming over the ocean from late autumn to early winter are common between the two periods, despite the difference in the source of the forcing (insolation vs. greenhouse gases).

(2) A chain of processes responsible for the warming trend from summer to autumn is elucidated by the decomposition to factors associated with sea surface temperature, ice concentration, and ice surface temperature changes.

(3) The downward clear-sky longwave radiation is one of major contributors to the model spread throughout the year. Other controlling terms vary with the season, but they are similar between the MH and the future in each season.

(4) The MH Arctic change may not be directly relevant to the future in some seasons (spring in particular) when the temperature response differs, but it is still useful to constrain the future Arctic projection (partly new addition to the original manuscript).

(5) The significant cross-model correlation found between summer albedo feedback and autumn-winter surface temperature response in both forcing cases suggests that feedbacks in preceding seasons, sea ice cover in particular, should not be overlooked as a constraint (new addition to the original manuscript).

[RC] A few general comments: * The analysis relies on four types of runs: Mid-Holocene (MH), Pre-industrial (PI), Historical and RCP4.5. I understand that MH simulations are taken from PMIP3, and so are PI simulations. I understand that Historical and RCP4.5 simulations are taken from CMIP5. Is that correct? Something confusing is that the authors write that "For the MH and PI simulations, we use monthly climatological data averaged over periods longer than a century, which were archived as part of the CMIP5 dataset" but also write that MH simulations were taken from PMIP3: "The MH simulation was designed and coordinated by the PMIP3 project". Could the authors clarify this at p. 2, line 30 (I did not find the explanations very clear).

[AC] We apologize for the confusion between PMIP3 and CMIP5. The MH experiment

was designed by PMIP3, and that was endorsed as a part of CMIP5. All the data were downloaded from CMIP5 data base. We will clarify this point.

[RC] * There is an important negative feedback that is not mentioned in the study: the negative ice growth-ice thickness feedback, which states that sea ice grows faster when it is thin. The existence of this feedback is a safeguard for sea ice, which would otherwise disappear much faster due to the positive albedo feedback. I'm unclear if the aforementioned negative feedback is covered at all by the authors and if so, to which term of Eq. 2 it belongs.

[AC] The negative ice growth-ice thickness feedback is not quantified explicitly in the current analysis. Therefore, it does not appear in the decomposed terms in Eq. (2) although they are closely linked to the sea ice related terms including the magnitude of albedo feedback (a function of ice cover among others) and heat release from the ocean (a function of ice thickness among others). Our analysis is based on the surface energy balance as in many other previous studies. The quantification of ice thickness feedback would require energy budget analysis for sea ice itself and probably for mixed-layer of the ocean as well. This does not mean that we think the feedback is unimportant. We will mention this point as a future perspective.

[RC] * There are two references missing that deal with high-latitude changes and the role of feedbacks, that I think should appear in the text: - DOI:10.1038/s41598-017-04623-7 - DOI: 10.1038/s41467-018-04173-0

[AC] Thank you for pointing out uncited references. We found these references useful and cite them in the revised manuscript.

[RC] Specific comments (Syntax: 22-03 = line 22, page 3) 19-01: "indirect atmospheric stratification" might be unclear to many. Please rephrase or explain.

[AC] We will rephrase the word.

[RC] 06-02: "time periods" –> "periods" (a period is always referring to time)

[AC] Will be corrected.

[RC] 07-02: "discouraged general comparisons": do you mean that the studies found that comparisons were not simple to make? Please rephrase.

[AC] We meant that the studies generally refuted to regard past warm periods as the analogue. We will rephrase it.

[RC] 23-02: "time periods" –> cf. 06-02.

[AC] Will be corrected.

[RC] 26-03: "effect" –> effects

[AC] We will change it as suggested.

[RC] 24-02: The last sentence of the paragraph is not quite clear; consider removing it.

[AC] We will remove it.

[RC] 22-04: "ts" shoud be T_s in mathematical form.

[AC] We will correct it.

[RC] 07-05: Why using the \Lambda sign for temperature differences, and not \Delta T? It is not clear how \Lambda relates to Eqs (6) and (4).

[AC] We will replace Lambda by T.

[RC] 28-05: Can you elaborate on how the ERF was computed precisely? It is said that an AGCM was used, but which one? What was the exact setup? It would be impossible to reproduce your results if the readers do not have this information.

[AC] The model information was only given in the figure caption. We will move this into the text, and also add more detailed description as to the setting.

---

## Author Comment (AC5) · 14 Mar 2019

Thank you very much for carefully reading the manuscript and for various helpful suggestions which would improve the manuscript and pointing out places that need to be clarified or discussed. In the following, reviewer's comments are indicated [RC]. Response to the comment and perspective on the revision are indicated by [AC].

[RC] General comment

This manuscript proposes an in depth analysis of the different radiative and turbulent latent and sensible heat fluxes terms that constraint the seasonal changes in surface temperature in the Arctic. The analysis considers the mid-Holocene climate and the RCP4.5 scenario for the future, with the objective to derive emerging constraints from

the mid-Holocene climate that can be used to assess the results of future climate projections. This analysis is interesting, but the conclusion is not strong enough about the analogies between the two periods and what can be done out of it. It is only during the ice melting period, when albedo decreases and water vapor increases in the atmosphere, that similar feedbacks occur. The forcing factors are very different between the two periods. Even though the different elements are found in the text, similarities and differences could be better discussed.

[AC] The objective of the current study is not to derive a specific emerging constraint but to reveal similarities and differences in processes, based on the detailed diagnosis, which are not obvious from the forcing and response patterns alone. The derivation of emerging constraints is one of ultimate goals beyond the scope of current study, and that would require several steps from statistical approach, mechanism understanding, and proxy searches. Even if similarities are found to be weaker or limited and they are unfavorable signs to find the specific emerging constraints in some cases, we do not think that would be fundamentally critical. The mechanism understanding of different periods under the same framework is really the most important aspect of the current study (which have not usually been done). For that, as the reviewer pointed out, it is also important to discuss differences between MH and future and the limitation of the use of MH climate information as well. As the differences were expected from the beginning due to different radiative forcing patters, we placed less emphasis on the differences. This might have led the impression that we were trying to stress the similarities. We will add more discussion on the differences.

[RC] The abstract could also be more informative on the results and better stress the role of the clear sky long wave radiation.

[AC] We will make the abstract (and conclusion) more specific so that the messages become clearer. The main points will be:

(1) It is found that many of the dominant processes that amplify Arctic warming over the

ocean from late autumn to early winter are common between the two periods, despite the difference in the source of the forcing (insolation vs. greenhouse gases).

(2) A chain of processes responsible for the warming trend from summer to autumn is elucidated by the decomposition to factors associated with sea surface temperature, ice concentration, and ice surface temperature changes.

(3) The downward clear-sky longwave radiation is one of major contributors to the model spread throughout the year. Other controlling terms vary with the season, but they are similar between the MH and the future in each season.

(4) The MH Arctic change may not be directly relevant to the future in some seasons (spring in particular) when the temperature response differs, but it is still useful to constrain the future Arctic projection (partly new addition to the original manuscript).

(5) The significant cross-model correlation found between summer albedo feedback and autumn-winter surface temperature response in both forcing cases suggests that feedbacks in preceding seasons, sea ice cover in particular, should not be overlooked as a constraint (new addition to the original mansucript).

[RC] The different figures are difficult to follow, because there is no direct relationship between the names of the different terms plotted in figure 5 (a key figure in this manuscript) and the decomposition done using equation 2 to 7. I therefore consider that this manuscript is worth publishing, but that an effort should be made to clarify the expression of the different terms and better explain their role.

[AC] We will change following points to improve the readability associated with Fig. 5 and equations.

(1) We will write terms in Eq. (4) explicitly after combining with Eq. (2), so that each term corresponds exactly to the description in Table 3 and each component in Figs. 5 and 10.

(2) We replace Lambda in Eq. (7) by T so that it is obvious that the symbol represents

temperature.

[RC] The discussion should also be enlarged, so that the paper more clearly address the point listed in the title.

[AC] The discussion will be substantially enlarged with separate points (1) in terms of the ensemble mean response, and (2) in terms of the model spread. We will also discuss not only the similarities but also for the difference between the MH and future (when and how). A particular attention will be paid to spring when the ensemble mean response differs between the two periods.

[RC] Other comments:

- P2 make sure you properly refer CMIP or PMIP everywhere.

[AC] We will revise the mixed use of terms, "CMIP" and "PMIP", and avoid confusion.

[RC] - P3 l 15. And section 4.3. The comparison of the MH results with observations is not fully used in the manuscript. Is there a way to go one step further by provided an evaluation that could really inform on the relevant processes between past and future?

[AC] It would be ideal to derive a specific emergent constraint and apply that to the model selection or to narrower quantitative uncertainty range, but the practical application is beyond the scope of the current paper. Nevertheless, we will add "recommendations" that the seasonal evolution of surface temperature response (cold season in practice) and likely summer sea ice cover are likely useful constraints based on the current analysis.

[RC] - P3 bl29. Could you provide an order of magnitude of the uncertainty related to emissivity for models that have a variable emissivity?

[AC] As shown by the good match of superimposed black solid and blue dashed lines in Fig. 5, the simulated temperature change and the sum of partial temperature changes calculated with unit emissivity are very similar for the ensemble mean, and also for

all individual models (not shown). For both annual and October-November-December means averaged over the Arctic ocean, for example, mismatches for all models are smaller than 0.06°C. Therefore, we think it is safe to assume the constant emissivity as in many previous studies.

[RC] - P4 2 Is the equation correct for S?

[AC] Thank you for catching this typographical error. In the second and third terms on the right side of Eq. (2), Delta alpha should be alpha, and S should be Delta S. The analysis was made correctly and the results are not affected. We will correct them.

[RC] - P4 l15 what do you call sect. 3a?

[AC] We will correct it to "Sect. 3.1".

[RC] - P4 end of section 3.1. It could be worth mentioning that the approach is direct because there is no change in land-sea mask between the different simulations.

[AC] Thank you for the suggestion. We will add this point

[RC] - P4. L 27 are your referring to ice concentration or to ice fraction?

[AC] As in the original manuscript, it is ice concentration, and it also represents fraction of ice cover for each grid cell. To clarify the procedure, we will add that the analysis of Eq. (6) is applied for each grid and each month.

[RC] - P5 Would it be possible to rewrite equation 7 so that there is a more direct link with temperature ? or use one example to fully explain what is done and the strength of the diagnosis. This could also be needed to present the different terms of equation (4) and make sure there is no ambiguity on global or local anomaly (or their relative strength).

[AC] As to the Eq. 7, we will replace the symbol Lambda by T so that it becomes clearer that the equation directly evaluates the temperature change and avoid any potential confusion. As written above, we will write terms in Eq. (4) explicitly after combining

with Eq. (2), so that each term corresponds exactly to the description in Table 3 and each component in Figs. 5 and 10.

[RC] - P4 l 8. May be you could site Hewitt and Mitchell 1997 for the definition of the MH insolation forcing.

[AC] It is nice to cite one of the earliest MH simulations in this context. We believe that it is Hewitt and Mitchell (1996, not 1997) in Journal of Climate. We will add it.

[RC] - P7 There is a large emphasize on clouds before showing the effect of lw_clr. This later term reflects both changes in water vapor and in atmospheric lapse rate. The cloud cld_effect arrives later (in season) compared to albedo and lw_clr. I would suggest reconsidering the way the whole section is written, to better discuss the relationship between the different terms and their monthly evolution.

[AC] In Sect. 4.3 in "Results", we will describe the results season by season first, and then state important points afterward so that the reader can grasp the overall results in the sequential order first.

[RC] - P8 section 4.5. I am not entirely convinced that OND are the best months to look at to infer model spread. Sea-ice and temperature result certainly of what happens during the preceding months in terms of forcing and feedbacks. This needs to be clarified.

[AC] We agree with the reviewer that the Arctic warming processes are not independent during each season. As briefly stated, we do not eliminate the possibility of links between feedbacks in other seasons and OND, for example: "this result does not mean that the summer albedo feedback is irrelevant to the OND model spread." Without numerical experiments, it is difficult to entangle the feedback links across seasons. We will discuss the inter-seasonal linkage between summer albedo feedback and OND response to the result which adds the important point.

[RC] - P9 section 4.5 I am lost in the call to the different figures.

[AC] We will add explanation on the relevance of these figures in the discussion here.

[RC] Figure 10 also show a large model spread in the lw_clr, not only in clouds. This should be highlighted. The cloud cover is important but results certainly from the other conditions: sea-ice fraction, temperature, lapse rate, water vapor, changes in atmospheric convection or large scale condensation. This should be discussed, at least to tell when there is an analogy or not between the different feedbacks between mid-Holocene and future climates.

[AC] We will stress the role of lw_clr throughout the paper. As stated in Sect. 4.5, the dominance of lw_clr was expected as most of downward longwave radiation comes from near-surface, and the near-surface temperature is thermally coupled with the surface temperature. Therefore, constraining lw_clr is equally difficult to constrain the surface temperature change. We will add the statement on lw_clr in abstract and conclusions so that the paper does not give false impression that the term is small or unimportant. We will mention that the cloud feedbacks are related to other feedbacks as pointed out by the reviewer. The discussion will be substantially enlarged with separate points (1) in terms of the ensemble mean response, and (2) in terms of the model spread. We will also discuss not only the similarities but also for the difference between the MH and future (when and how). A particular attention will be paid to spring when the ensemble mean response differs between the two periods but factors for the model spread are similar. Consequently, as the reviewer suggests, discussion on the existence and non-existence of analogy in feedbacks between MH and future for different seasons will be added.

---

## Author Response (AR1)

**Replies to comment by anonymous referee #1**

Thank you very much for carefully reading the manuscript and for bringing up some important points. In the following, reviewer's comments are indicated by [RC]. Response to the comment and changes in the manuscript are indicated by [AC]. MS stands for manuscript.

[RC] This paper discusses the decomposition of the changes in surface temperature into local and global feedback contributions, related to the different components of the surface energy balance. This decomposition is performed both for Mid-Holocene (compared to pre-industrial) and future warming (under RCP4.5 scenario).

As a general comment I would say that I found the paper difficult to read, although I cannot figure out exactly the reason (either the topic or the language).

[AC] Taking also the suggestions by other reviewers into account, we revised following points to improve the readability.

(1) We changed the abstract to be more informative with emphasis on the specific new findings.

(2) We wrote terms in Eq. (4) explicitly after combining with Eq. (2), so that each term corresponds exactly to the description in Table 3 and each component in Figs. 5 and 10.

(3) We replaced Lambda in Eq. (7) by T so that it is obvious that the symbol represents temperature.

(4) In Sect. 4.3 in "Results", we describe the results season by season first, and then state important points afterward so that the reader can grasp the overall results in the sequential order in the revised MS.

(5) In Sect. 4.5 in "Results", we describe the results season by season first, and then state important points afterward for the same reason as (4) in the revised MS.

[RC] I also found that the discussion is not really a discussion, but rather a perspective and conclusion. After reading the manuscript, and although I acknowledge similarities in climate changes between MH and the future, I still do not understand the 'relevance of mid-Holocene Arctic warming to the future '. This should be the major item in the discussion.

[AC] We enlarged the discussion and conclusion with emphasis on the relevance in the Arctic response between the MH and future (RCP4.5). The discussion was substantially enlarged with separate points (1) in terms of the ensemble mean response, and (2) in terms of the model spread. We also increased discussion for the difference between the MH and future (when and how).

[RC] The conclusion of the paper is not very new. It has already been repeated many times that

'improvement of the ability of the model to simulate the past will increase the confidence in their ability to simulate the future'. I would suggest to identify a 'crispier' conclusion.

[AC] We made the conclusion more specific and removed general statements from the conclusion.

The main points are:

(1) It is found that many of the dominant processes that amplify Arctic warming over the ocean from late autumn to early winter are common between the two periods, despite the difference in the source of the forcing (insolation vs. greenhouse gases).

(2) A chain of processes responsible for the warming trend from summer to autumn is elucidated by the decomposition to factors associated with sea surface temperature, ice concentration, and ice surface temperature changes.

(3) The downward clear-sky longwave radiation is one of major contributors to the model spread throughout the year. Other controlling terms vary with the season, but they are similar between the MH and the future in each season.

(4) The MH Arctic change may not be analogous to the future in some seasons (spring in particular) when the temperature response differs, but it is still useful to constrain the future Arctic projection.

(5) The significant cross-model correlation found between summer albedo feedback and autumn- winter surface temperature response in both forcing cases suggests that feedbacks in preceding seasons, sea ice cover in particular, should not be overlooked as a constraint.

[RC] The manuscript is missing a data availability section. Moreover, data citations are also missing (in addition to the references that are indeed given). This is the case, at least for the data of Bartlein et al (2011) and Sundqvist et al (2010). Moreover, the code used to extract the values displayed in figure 5 should be made publicly available as well (with a reference in the data availability section).

[AC] We added Data Availability section. We also make the computer codes used for the analysis in Figs. 5 (, 7, and 10) available upon acceptance of the paper and upon request.

[RC] Specific comments.

P1-l29 : is it solar forcing?

[AC] Marshall et al. (2014) suggests stratospheric ozone forcing. To be precise, we changed it to

"stratospheric ozone change and cloud feedback play additional roles".

[RC] P2-l1 : I assume that the 'scenario' refers to RCP scenarios. This should be made clear.

[AC] Change was made to "RCP scenario".

[RC] P4-l16 and P7-l13 : there is a reference to Sect. 3a, which does not exist (at least as such).

[AC] "3a" should be "3.1". It was corrected.

[RC] P6-l33 : According to my reading of the figure, the simulated warming only occurs in the northern North Atlantic and Arctic oceans, where there is no data. It is therefore very difficult to say if it is under- or over-estimated. Or do the authors call 'warming' the negative values in the figure?

[AC] We changed it to "the warming indicated by the reconstruction is not captured by the model mean in January as well as in the annual mean."

[RC] P7-l16 : 'plays an important role'. According to my reading, this is only true in JJA.

[AC] We moved the corresponding sentence to the description for JJA.

[RC] P7-l32 : 'exhibits a large contribution'. This does not really seem to be the case for MH.

[AC] The reviewer is correct. We made distinction between RCP4.5 and MH in the revised MS.

[RC] P8-l9 : could the authors make the label coherent (Dtas in the text, Dta in the figure).

[AC] The correction was made to the text.

[RC] P10-l11 : PMIP3 instead of PMIM3

[AC] Corrected. Thank you.

[RC] P10-l22 : The authors should make their conclusion readable by itself. It should be said that the Arctic warming is for the future (under RCP4.5).

[AC] We made the conclusion readable by itself by adding words in the revised MS.

[RC] P10-l33 : 'seeking possible analogues between physical processes in the past and future climate'. Do the authors mean that the climate processes are time dependent? I thought that they were based on basic physical principles valid through time. Moreover, as we do not know the future climate it is hard to look for analogues there and then.

[AC] While physical principles are same throughout the time, what we meant is, it is not trivial that the dominating processes for the climate variations are the same for different climate forcing and change cases. We rephrased the sentence.

[RC] P15 : A reference is missing here.

[AC] We added the reference.

[RC] P 21 : The figure is misleading because the Y-axis (scale) is not the same for MH and RCP4.5.

[AC] We added the note on the caption. The difference in magnitude does not preclude the use of these two different time periods, and rather it is of interest that such different climate responses still share the similar dominant processes.

[RC] P23-l4 : I do not see two (black and blue dashed) lines. Are they exactly superimposed? In that case, this should be mentioned in the caption.

[AC] They are black polygonal solid-line and blue polygonal dashed-line. We made the caption more precise (and text) in the revised MS.

[RC] P24-26 : Figures 6-8 are not using the same number of models. (1) the name of the models used should be mentioned. (2) Not using all the models (and not always the same models) may introduce a bias in the interpretation. Would the conclusion be the same if only the models (and their outputs) available for all the figures were used?

[AC]

(1)  All model names were given in Table 2, but the models used for Fig. 7 was only written in text. We will refer Table 2 for Figs. 6 and 8, and write names explicitly for Fig. 7 in the caption.

(2)  The main results shown in Figs. 5 and 10 are benefitted the most by using the models as many as possible (10 models), but all variables are available only for 5 models: 5 models are missing for Fig. 7 and 1 model is missing for Fig. 8 (It was mistakenly written that 2 models were missing in the caption of Fig. 8. We corrected it). We checked the consistency of Figs. 5, 7

and 8 by reducing the model numbers to 5. Figs. 5 and 8 were not qualitatively affected by this reduction. We also checked all figures by reducing the model numbers to 5: a few small terms lost their statistical significance in Fig. 10, but the conclusion remains the same.

**Replies to comment by referee #2 (Dr. Massonnet)**

Thank you very much for carefully reading the manuscript and for pointing out some of the messages that need to be sharpened. In the following, reviewer's comments are indicated by [RC]. Response to the comment and changes in the manuscript are indicated by [AC]. MS stands for manuscript.

[RC] In this study, the authors conduct a diagnostic surface balance analysis based on output from the PMIP3 and CMIP5 simulations. They wish to test the extent to which past Arctic warming could be used as an analogue for future warming, which could then make the basis for a more objective model selection. The authors find that despite different forcing mechanisms, several common feedbacks operate between the two periods, making the case that these periods can indeed be compared to one another.

The paper is interesting to read but is quite descriptive: the differences between the MH-PI and RCP4.5-Historical simulations are stated and described, but not a lot of attention is given to try to explain why patterns differ and, more importantly, why this would have implications for the scientific community.

[AC] In the original manuscript, the similarity in feedbacks in particular season (autumn) might have been too emphasized, and less attention was paid to the difference between the MH and future. We enlarged the discussion and conclusion with emphasis on the relevance in the Arctic response between the MH and future (RCP4.5). The discussion was substantially enlarged with separate points (1) in terms of the ensemble mean response, and (2) in terms of the model spread. In the revised MS, we also discuss not only the similarities but also for the difference between the MH and future (when and how). A particular attention was paid to spring when the ensemble mean response differs between the two periods. In addition, we increased quantitative description.

[RC] The conclusions fall a bit short, for example. The authors explain that the MH period could be used to evaluate the models, but they do not state what type of constraints could be applied. Based on their results, can the authors make a step forward and come with recommendations on such constraints?

[AC] We do not claim any new 'emergent constraints' in the current study although that would offer more practical implication. We believe that the application of such constraint should go hand in hand with mechanism understanding, statistical identification of the link between the past and the future (e.g., Schmidt et al., 2014), and paleoclimate proxy searches suitable to constrain the link. Nevertheless, in the revised MS, we add "recommendations" that the seasonal evolution of surface temperature response (cold season in practice) and likely summer sea ice cover are likely useful constraints based on the current analysis. In addition, we made the conclusion (and abstract) more specific so that the messages become clearer. The main points are:

(1) It is found that many of the dominant processes that amplify Arctic warming over the ocean from late autumn to early winter are common between the two periods, despite the difference in the source of the forcing (insolation vs. greenhouse gases).

(2) A chain of processes responsible for the warming trend from summer to autumn is elucidated by the decomposition to factors associated with sea surface temperature, ice concentration, and ice surface temperature changes.

(3) The downward clear-sky longwave radiation is one of major contributors to the model spread throughout the year. Other controlling terms vary with the season, but they are similar between the MH and the future in each season.

(4) The MH Arctic change may not be analogous to the future in some seasons (spring in particular) when the temperature response differs, but it is still useful to constrain the future Arctic projection.

(5) The significant cross-model correlation found between summer albedo feedback and autumn- winter surface temperature response in both forcing cases suggests that feedbacks in preceding seasons, sea ice cover in particular, should not be overlooked as a constraint.

[RC] A few general comments:

* The analysis relies on four types of runs: Mid-Holocene (MH), Pre-industrial (PI), Historical and RCP4.5. I understand that MH simulations are taken from PMIP3, and so are PI simulations.

I understand that Historical and RCP4.5 simulations are taken from CMIP5. Is that correct?

Something confusing is that the authors write that "For the MH and PI simulations, we use monthly climatological data averaged over periods longer than a century, which were archived as part of the CMIP5 dataset" but also write that MH simulations were taken from PMIP3: "The MH

simulation was designed and coordinated by the PMIP3 project". Could the authors clarify this at p. 2, line 30 (I did not find the explanations very clear).

[AC] We apologize for the confusion between PMIP3 and CMIP5. The MH experiment was designed by PMIP3, and that was endorsed as a part of CMIP5. All the data were downloaded from CMIP5 data base. We clarified this point in the revised MS.

[RC] * There is an important negative feedback that is not mentioned in the study: the negative ice growth-ice thickness feedback, which states that sea ice grows faster when it is thin. The existence of this feedback is a safeguard for sea ice, which would otherwise disappear much faster due to the positive albedo feedback. I'm unclear if the aforementioned negative feedback is covered at all by the authors and if so, to which term of Eq. 2 it belongs.

[AC] The negative ice growth-ice thickness feedback is not quantified explicitly in the current analysis. Therefore, it does not appear in the decomposed terms in Eq. (2) although they are closely linked to the sea ice related terms including the magnitude of albedo feedback (a function of ice cover among others) and heat release from the ocean (a function of ice thickness among others). Our analysis is based on the surface energy balance as in many other previous studies.

The quantification of ice thickness feedback would require energy budget analysis for sea ice itself and probably for mixed-layer of the ocean as well. This does not mean that we think the feedback is unimportant. We mention this point as a future perspective in the revised MS.

[RC] * There are two references missing that deal with high-latitude changes and the role of feedbacks, that I think should appear in the text:

- DOI:10.1038/s41598-017-04623-7

- DOI: 10.1038/s41467-018-04173-0

[AC] Thank you for pointing out uncited references. We found these references useful and cite them in the revised MS.

[RC] Specific comments (Syntax: 22-03 = line 22, page 3)

19-01: "indirect atmospheric stratification" might be unclear to many. Please rephrase or explain.

[AC] "indirect" was removed.

[RC] 06-02: "time periods" –> "periods" (a period is always referring to time)

[AC] Corrected.

[RC] 07-02: "discouraged general comparisons": do you mean that the studies found that comparisons were not simple to make? Please rephrase.

[AC] We meant that the studies generally refuted to regard past warm periods as the analogue.

We rewrote it.

[RC] 23-02: "time periods" –> cf. 06-02.

[AC] Corrected.

[RC] 26-03: "effect" –> effects

[AC] We changed it as suggested.

[RC] 24-02: The last sentence of the paragraph is not quite clear; consider removing it.

[AC] We removed it.

[RC] 22-04: "ts" shoud be T_s in mathematical form.

[AC] We corrected it.

[RC] 07-05: Why using the ¥Lambda sign for temperature differences, and not ¥Delta T? It is not clear how ¥Lambda relates to Eqs (6) and (4).

[AC] We replaced Lambda by T.

[RC] 28-05: Can you elaborate on how the ERF was computed precisely? It is said that an AGCM

was used, but which one? What was the exact setup? It would be impossible to reproduce your results if the readers do not have this information.

[AC] The model information was only given in the figure caption. We moved this into the text, and also added more detailed description as to the setting in the revised MS.

**Replies to comment by anonymous referee #3**

Thank you very much for carefully reading the manuscript and for various helpful suggestions which would improve the manuscript and pointing out places that need to be clarified or discussed. In the following, reviewer's comments are indicated [RC]. Response to the comment and changes in the revision are indicated by [AC]. MS stands for manuscript.

[RC] General comment

This manuscript proposes an in depth analysis of the different radiative and turbulent latent and sensible heat fluxes terms that constraint the seasonal changes in surface temperature in the Arctic. The analysis considers the mid-Holocene climate and the RCP4.5 scenario for the future, with the objective to derive emerging constraints from the mid-Holocene climate that can be used to assess the results of future climate projections. This analysis is interesting, but the conclusion is not strong enough about the analogies between the two periods and what can be done out of it. It is only during the ice melting period, when albedo decreases and water vapor increases in the atmosphere, that similar feedbacks occur. The forcing factors are very different between the two periods. Even though the different elements are found in the text, similarities and differences could be better discussed.

[AC] The objective of the current study is not to derive a specific emerging constraint but to reveal similarities and differences in processes, based on the detailed diagnosis, which are not obvious from the forcing and response patterns alone. The derivation of emerging constraints is one of ultimate goals beyond the scope of the current study, and that would require several steps from statistical approach, mechanism understanding, and proxy searches. Even if similarities are found to be weaker or limited and they are unfavorable signs to find the specific emerging constraints in some cases, we do not think that would be fundamentally critical. The mechanism understanding of different periods under the same framework is really the most important aspect of the current study (which have not usually been done). For that, as the reviewer pointed out, it is also important to discuss differences between MH and future and the limitation of the use of MH climate information as well. As the differences were expected from the beginning due to different radiative forcing patters, we placed less emphasis on the differences. This might have led the impression that we were trying to stress the similarities. Therefore, we added more discussion on the differences in the revised MS.

[RC] The abstract could also be more informative on the results and better stress the role of the clear sky long wave radiation.

[AC] We made the abstract (and conclusion) more specific so that the messages become clearer.

The main points are:

(1) It is found that many of the dominant processes that amplify Arctic warming over the ocean from late autumn to early winter are common between the two periods, despite the difference in the source of the forcing (insolation vs. greenhouse gases).

(2) A chain of processes responsible for the warming trend from summer to autumn is elucidated by the decomposition to factors associated with sea surface temperature, ice concentration, and ice surface temperature changes.

(3) The downward clear-sky longwave radiation is one of major contributors to the model spread throughout the year. Other controlling terms vary with the season, but they are similar between the MH and the future in each season.

(4) The MH Arctic change may not be analogous to the future in some seasons (spring in particular) when the temperature response differs, but it is still useful to constrain the future Arctic projection.

(5) The significant cross-model correlation found between summer albedo feedback and autumn- winter surface temperature response in both forcing cases suggests that feedbacks in preceding seasons, sea ice cover in particular, should not be overlooked as a constraint.

[RC] The different figures are difficult to follow, because there is no direct relationship between the names of the different terms plotted in figure 5 (a key figure in this manuscript) and the decomposition done using equation 2 to 7. I therefore consider that this manuscript is worth publishing, but that an effort should be made to clarify the expression of the different terms and better explain their role.

[AC] We changed following points to improve the readability associated with Fig. 5 and equations.

(1) We wrote terms in Eq. (4) explicitly after combining with Eq. (2), so that each term corresponds exactly to the description in Table 3 and each component in Figs. 5 and 10.

(2) We replaced Lambda in Eq. (7) by T so that it is obvious that the symbol represents temperature.

[RC] The discussion should also be enlarged, so that the paper more clearly address the point listed in the title.

[AC] The discussion was substantially enlarged with separate points (1) in terms of the ensemble mean response, and (2) in terms of the model spread. In the revised MS, We also discuss not only
the similarities but also for the difference between the MH and future (when and how). A
particular attention was paid to spring when the ensemble mean response differs between the two
periods.
[RC] Other comments:
- P2 make sure you properly refer CMIP or PMIP everywhere.
[AC] We revised the mixed use of terms, "CMIP" and "PMIP" to avoid confusion.
[RC] - P3 l 15. And section 4.3. The comparison of the MH results with observations is not fully
used in the manuscript. Is there a way to go one step further by provided an evaluation that could
really inform on the relevant processes between past and future?
[AC] It would be ideal to derive a specific emergent constraint and apply that to the model
selection or to narrower quantitative uncertainty range, but the practical application is beyond the
scope of the current paper. Nevertheless, in the revised MS, we added "recommendations" that
the seasonal evolution of surface temperature response (cold season in practice) and likely
summer sea ice cover are likely useful constraints based on the current analysis.
[RC] - P3 bl29. Could you provide an order of magnitude of the uncertainty related to emissivity
for models that have a variable emissivity?
[AC] As shown by the good match of superimposed black solid and blue dashed lines in Fig. 5,
the simulated temperature change and the sum of partial temperature changes calculated with unit
emissivity are very similar for the ensemble mean, and also for all individual models (not shown).
For both annual and October-November-December means averaged over the Arctic ocean, for
example, mismatches for all models are smaller than 0.06°C. Therefore, it is safe to assume the
constant emissivity as in many previous studies.
[RC] - P4 2 Is the equation correct for S?
[AC] Thank you for catching this typographical error. In the second and third terms on the right
side of Eq. (2), Delta alpha should be alpha, and S should be Delta S. The analysis was made
correctly and the results are not affected. We corrected them in the revised MS.

[RC] - P4 l15 what do you call sect. 3a?

[AC] We corrected it to "Sect. 3.1".

[RC] - P4 end of section 3.1. It could be worth mentioning that the approach is direct because there is no change in land-sea mask between the different simulations.

[AC] Thank you for the suggestion. We added this point at the place suggested.

[RC] - P4. L 27 are your referring to ice concentration or to ice fraction?

[AC] As in the original manuscript, it is ice concentration, and it also represents fraction of ice cover for each grid cell. To clarify the procedure, we added the remark that the analysis of Eq. (6)

is applied for each grid and each month in the revised MS.

[RC] - P5 Would it be possible to rewrite equation 7 so that there is a more direct link with temperature ? or use one example to fully explain what is done and the strength of the diagnosis.

This could also be needed to present the different terms of equation (4) and make sure there is no ambiguity on global or local anomaly (or their relative strength).

[AC] As to Eq. 7, we replaced the symbol $\Lambda$ by $\Delta T$ so that it becomes clearer that the equation directly evaluates the temperature change $\Delta T$ and avoid any potential confusion. As written above, we wrote terms in Eq. (4) explicitly after combining with Eq. (2) in the revised MS, so that each term corresponds exactly to the description in Table 3 and each component in Figs. 5 and 10.

[RC] - P4 l 8. May be you could site Hewitt and Mitchell 1997 for the definition of the MH

insolation forcing.

[AC] It is nice to cite one of the earliest MH simulations in this context. We believe that it is

Hewitt and Mitchell (1996, not 1997) in Journal of Climate. We added it.

[RC] - P7 There is a large emphasize on clouds before showing the effect of lw_clr. This later term reflects both changes in water vapor and in atmospheric lapse rate. The cloud cld_effect arrives later (in season) compared to albedo and lw_clr. I would suggest reconsidering the way the whole section is written, to better discuss the relationship between the different terms and their monthly evolution.

[AC] In Sect. 4.3 in "Results", we describe the results season by season first, and then state important points afterward so that the reader can grasp the overall results in the sequential order first in the revised MS.

[RC] - P8 section 4.5. I am not entirely convinced that OND are the best months to look at to infer model spread. Sea-ice and temperature result certainly of what happens during the preceding months in terms of forcing and feedbacks. This needs to be clarified.

[AC] We agree with the reviewer that the Arctic warming processes are not independent during each season. As briefly stated, we do not eliminate the possibility of links between feedbacks in other seasons and OND, for example: "this result does not mean that the summer albedo feedback is irrelevant to the OND model spread." Without numerical experiments, it is difficult to entangle the feedback links across seasons. In the revised MS, we discuss the inter-seasonal linkage between summer albedo feedback and OND response which adds the important point.

[RC] - P9 section 4.5 I am lost in the call to the different figures.

[AC] In the revised MS, we add explanation on the relevance of these figures in the discussion here.

[RC] Figure 10 also show a large model spread in the lw_clr, not only in clouds. This should be highlighted. The cloud cover is important but results certainly from the other conditions: sea-ice fraction, temperature, lapse rate, water vapor, changes in atmospheric convection or large scale condensation. This should be discussed, at least to tell when there is an analogy or not between the different feedbacks between mid-Holocene and future climates.

[AC] In the revised MS, we stress the role of lw_clr throughout the paper. As stated in Sect. 4.5, the dominance of lw_clr was expected as most of downward longwave radiation comes from near- surface, and the near-surface temperature is thermally coupled with the surface temperature.

Therefore, constraining lw_clr is equally difficult to constrain the surface temperature change.

We added the statement on lw_clr in abstract and conclusions so that the paper does not give false impression that the term is small or unimportant. In the revised MS, we mention that the cloud feedbacks are related to other feedbacks as pointed out by the reviewer. The discussion will be substantially enlarged with separate points (1) in terms of the ensemble mean response, and (2)

in terms of the model spread. In the revised MS, we also discuss not only the similarities but also for the difference between the MH and future (when and how). A particular attention was paid to spring when the ensemble mean response differs between the two periods but factors for the model spread are similar. Consequently, as the reviewer suggests, discussion on the existence and non- existence of analogy in feedbacks between MH and future for different seasons were added in the revised MS.

[revised manuscript text omitted]

**Page 17: [1] Formatted**                                    **Author**

Font color: Text 1

**Page 17: [2] Formatted**                                    **Author**

Font color: Text 1, English (UK)

**Page 17: [3] Formatted**                                    **Author**

EndNote Bibliography, Indent: Left:  0 cm, Hanging:  1.27 cm, First line:  0 ch

**Page 17: [4] Formatted**                                    **Author**

Font color: Text 1, English (UK)

**Page 17: [5] Formatted**                                    **Author**

Font: Not Italic, Font color: Text 1, English (UK)

**Page 17: [5] Formatted**                                    **Author**

Font: Not Italic, Font color: Text 1, English (UK)

**Page 17: [6] Formatted**                                    **Author**

Font color: Text 1, English (UK)

**Page 17: [7] Deleted**                                    **Author**

**Page 17: [8] Formatted**                                    **Author**

Font color: Text 1, English (UK)

**Page 17: [9] Deleted**                                    **Author**

**Page 17: [10] Formatted**                                    **Author**

Font: Not Italic, Font color: Text 1, English (UK)

**Page 17: [11] Deleted**                                    **Author**

**Page 17: [12] Formatted**                                    **Author**

Font color: Text 1, English (UK)

**Page 17: [13] Deleted**                                    **Author**

**Page 17: [14] Formatted**                                    **Author**

Font color: Text 1, English (UK)

**Page 17: [15] Deleted**                                    **Author**

**Page 17: [16] Formatted**                                    **Author**

Font: Not Italic, Font color: Text 1, English (UK)

| Page 17: [17] Deleted | Author |
|---|---|

| Page 17: [18] Formatted | Author |
|---|---|

Font: Not Italic, Font color: Text 1, English (UK)

| Page 17: [18] Formatted | Author |
|---|---|

Font: Not Italic, Font color: Text 1, English (UK)

| Page 17: [19] Deleted | Author |
|---|---|

| Page 17: [20] Formatted | Author |
|---|---|

Font color: Text 1, English (UK)

| Page 17: [21] Deleted | Author |
|---|---|

| Page 17: [22] Formatted | Author |
|---|---|

Font color: Text 1, German

| Page 17: [23] Deleted | Author |
|---|---|

| Page 17: [24] Formatted | Author |
|---|---|

Font: Italic, Font color: Text 1, German

| Page 17: [25] Deleted | Author |
|---|---|

| Page 17: [26] Formatted | Author |
|---|---|

Font color: Text 1, German

| Page 17: [27] Deleted | Author |
|---|---|

| Page 17: [28] Formatted | Author |
|---|---|

Font color: Text 1

| Page 17: [29] Deleted | Author |
|---|---|

| Page 17: [30] Formatted | Author |
|---|---|

Font color: Text 1

| Page 17: [31] Formatted | Author |
|---|---|

Font color: Text 1

| Page 17: [32] Formatted | Author |
|---|---|

Font color: Text 1

Font color: Text 1
* * *
**Page 17: [34] Formatted**                                **Author**

Font color: Text 1
* * *
**Page 17: [35] Formatted**                                **Author**

Font color: Text 1
* * *
**Page 17: [36] Formatted**                                **Author**

Font color: Text 1
* * *
**Page 17: [37] Formatted**                                **Author**

Font color: Text 1
* * *
**Page 17: [38] Formatted**                                **Author**

Font color: Text 1
* * *
**Page 17: [39] Formatted**                                **Author**

Font color: Text 1
* * *
**Page 17: [40] Formatted**                                **Author**

Font color: Text 1
* * *
**Page 17: [41] Formatted**                                **Author**

Font color: Text 1
* * *
**Page 17: [42] Formatted**                                **Author**

Font color: Text 1
* * *
**Page 17: [43] Formatted**                                **Author**

Font color: Text 1
* * *
**Page 17: [44] Formatted**                                **Author**

Font color: Text 1
* * *
**Page 17: [45] Formatted**                                **Author**

Font color: Text 1
* * *
**Page 17: [46] Formatted**                                **Author**

Font color: Text 1
* * *
**Page 17: [47] Formatted**                                **Author**

Font color: Text 1
* * *
**Page 17: [48] Formatted**                                **Author**

Font color: Text 1

**Page 17: [50] Formatted**        **Author**

Font color: Text 1

**Page 17: [51] Formatted**        **Author**

Font color: Text 1

**Page 17: [52] Formatted**        **Author**

Font color: Text 1

**Page 17: [53] Formatted**        **Author**

Font color: Text 1

**Page 17: [54] Formatted**        **Author**

Font color: Text 1

**Page 17: [55] Formatted**        **Author**

Font color: Text 1

**Page 17: [56] Formatted**        **Author**

Font color: Text 1

**Page 17: [57] Formatted**        **Author**

Font color: Text 1

**Page 17: [58] Formatted**        **Author**

Font color: Text 1

**Page 17: [59] Deleted**        **Author**

**Page 17: [60] Deleted**        **Author**

**Page 17: [61] Deleted**        **Author**

**Page 17: [62] Deleted**        **Author**

**Page 17: [63] Deleted**        **Author**

**Page 17: [64] Deleted**        **Author**

**Page 17: [65] Deleted**        **Author**

**Page 17: [66] Deleted**        **Author**

**Page 17: [67] Formatted**        **Author**

English (US)

**Page 17: [68] Deleted**        **Author**

English (US)

| Page 17: [70] Deleted | Author |
|---|---|

| Page 17: [71] Formatted | Author |
|---|---|

Font color: Text 1

| Page 17: [72] Formatted | Author |
|---|---|

Font color: Text 1

| Page 18: [73] Deleted | Author |
|---|---|

| Page 18: [74] Deleted | Author |
|---|---|

| Page 18: [75] Deleted | Author |
|---|---|

| Page 18: [76] Deleted | Author |
|---|---|

| Page 18: [77] Deleted | Author |
|---|---|

| Page 18: [78] Deleted | Author |
|---|---|

| Page 18: [79] Deleted | Author |
|---|---|

| Page 18: [80] Deleted | Author |
|---|---|

| Page 18: [81] Deleted | Author |
|---|---|

| Page 19: [82] Deleted | Author |
|---|---|

**The relevance of mid-Holocene Arctic warming to the future**

**(Supplementary table and figures)**

Masakazu Yoshimori[1,2*] and Marina Suzuki[3]

[1] Faculty of Environmental Earth Science, Global Institution for Collaborative Research
and Education, and Arctic Research Center,
Hokkaido University, Sapporo, Japan

[2] Atmosphere and Ocean Research Institute, The University of Tokyo, Kashiwa, Japan

[3] Graduate School of Environmental Science, Hokkaido University, Sapporo, Japan

Manuscript is to be submitted to *Climate of the Past* (24 April 2019)

* Corresponding author address: Atmosphere and Ocean Research Institute, The
University of Tokyo, 5-1-5, Kashiwanoha, Kashiwa, Chiba 277-8568 Japan
Phone: +81-4-7136-4380, Fax: +81-4-7136-4375, E-mail: masakazu@aori.u-tokyo.ac.jp

Table S1    Model years used to construct the long-term climatology for the PI and MH simulations from

              "r1i1p1" runs in the CMIP5 dataset.

| Model | PI | MH |
|---|---|---|
| bcc-csm1-1 | 0001 - 0500 | 0001 - 0100 |
| CCSM4 | 0800 - 1300 | 1000 - 1300 |
| CNRM-CM5 | 1850 - 2699 | 1950 - 2149 |
| CSIRO-Mk3-6-0 | 0001 - 0500 | 0001 - 0100 |
| FGOALS-g2 | 0001 - 0900 | 3400 - 1024 |
| FGOALS-s2 | 1850 - 2350 | 0001 - 0100 |
| GISS-E2-R | 3331 - 4530 | 2500 - 2599 |
| IPSL-CM5A-LR | 2370 - 2799 | 2710 - 2800 |
| MIROC-ESM | 1800 - 2429 | 2330 - 2429 |
| MRI-CGCM3 | 1851 - 2350 | 1951 - 2050 |

___________________________

[Figure]

Fig. S1   Seasonal progress of the zonal mean radiative forcing calculated with the insolation anomaly for ΔMH and planetary albedo from the PI experiment (W m$^{-2}$). The mean of all 10 models was used. See main text for details.

[Figure]

Fig. S2   Surface air temperature anomaly (°C) for ΔMH from the reconstruction (left) and simulations (right): (a) & (b) annual mean, (c) & (d) warmest month, and (e) & (f) coldest month. The reconstruction data are taken from the extended data of Bartlein et al. (2011). The mean of all 10 model simulations was used.

[Figure]

[Figure]

Figure S3 Simulated and diagnosed surface temperature changes (°C) for the land (north of 60°N): (a) ΔMH; and (b) ΔRCP4.5. The black polygonal solid lines denote simulated changes and blue polygonal dashed lines denote the sum of diagnosed partial changes; two lines are superimposed. The graphs represent the means of all 10 models listed in Table 2. See Table 3 for the interpretation of each component.

[Figure]

[Figure]

Figure S4 Fractional contribution of individual processes to the simulated surface temperature change (%) over the land (north of 60°N) for ΔMH and ΔRCP4.5: (a) spring (March-April-May); (b) summer (June-

July-August); (c) autumn (September-October-November); and (d) winter (December-January-February)

means. The sum of the bar graphs in the same color for each plot adds up to 100%. The hatching indicates the contribution is statistically significant at the 10% level. All 10 models listed in Table 2 are used. See

Table 3 for the interpretation of each component. Note that the vertical scale for (b) is three times larger than others.

**Reference**

Bartlein, P. J., and Coauthors: Pollen-based continental climate reconstructions at 6 and 21 ka: a global synthesis. *Climate Dynamics*, **37,** 775-802. DOI 10.1007/s00382-010-0904-1, 2011

---

## Author Response (AR2)

**Replies to comment by anonymous referee #3**

Thank you very much for reviewing the manuscript and additional comments that help further improve the manuscript. In the following, reviewer's comments are indicated by [RC]. Response to the comment and changes in the manuscript are indicated by [AC].

[RC] The new version of the manuscript shows substantial improvements compared to the initial version. It is now quite complete and the discussion is more interesting and focused. The authors considered almost all my comments. I still have minor comments that could make the manuscript even more complete. I understand that the authors do not want to address the question of emerging constraint and that they restrict the manuscript to the understanding. I fully agree with this point of view. Then adding a figure on the atmospheric column to show the changes in lapse rate and humidity (specific or relative depending on what seems more appropriate) would be great and add to the discussion.

Other minor comments:
The abstract could be slightly shortened to make sure the key messages appear strong enough.
How the simulations are combined and which variables are considered to compute the different terms in equation 5 is not entirely obvious, and a few lines explaining this important aspect in more details would be welcome. If possible, in addition to figure 6-9 one or 2 figures showing the seasonal evolution of the change in atmospheric temperature and humidity would help understand the polar amplification, as well as differences in cloud and in some of the surface fluxes.

[AC]
1. As suggested, we added two figures of seasonal progress of anomalous air temperature and specific humidity profiles as Figs. 7 and 8 accompanied by some discussions. They are actually helpful. The corresponding sentences are as follows. For the model ensemble mean: "This term includes the effect of air temperature and specific humidity changes (and also the radiative forcing of greenhouse gases for the case of $\Delta$RCP4.5), and is qualitatively consistent with changes in both variables (Figs. 7a and 7c, and Figs. 8a and 8c)." For the model spread: "The model variances of the air temperature change are concentrated near the surface in non-summer season (Figs. 7b and 7d), and those of the specific humidity change are large in non-spring season (Figs. 8b and 8d). The relative contribution of air temperature and water vapor to the clear-sky LW radiation may thus vary with the season.". For the interpretation of $\Delta$MH response: "In addition, the peak mid-tropospheric cooling in spring and warming in summer for $\Delta$MH in Fig. 7a are suggestive of remote influence through atmospheric heat transport."

As the surface fluxes are shown in terms of partial surface temperature change unit in Fig. 5, we choose not to add further figures (also to keep the number of figures in a reasonable size; individual radiation fluxes, sensible, and latent fluxes). The cloud profile is of interest, but the main message here is to point out the importance of cloud radiation feedback both for the ensemble mean and variance, for which we think the total cloud fraction already shown in Fig. 6 is sufficient (rather than going into details of complex cloud response), and thus is not added.

2. We have shortened the abstract (363 to 320 words). Indeed, it became concise and is in better shape now, we think. Thank you for the suggestion.

[revised manuscript text omitted]

**Page 17: [1] Formatted**                                    **Author**

Left, Indent: Left:  0 cm, Hanging:  0.75 cm

**Page 17: [2] Deleted**                                      **Author**

**Page 17: [2] Deleted**                                      **Author**

**Page 17: [2] Deleted**                                      **Author**

**Page 17: [2] Deleted**                                      **Author**

**Page 17: [2] Deleted**                                      **Author**

**Page 17: [3] Formatted**                                    **Author**

Font: Not Italic

**Page 17: [4] Formatted**                                    **Author**

Font: Not Bold

**Page 17: [5] Deleted**                                      **Author**

**Page 17: [5] Deleted**                                      **Author**

**Page 17: [5] Deleted**                                      **Author**

**Page 17: [5] Deleted**                                      **Author**

**Page 17: [5] Deleted**                                      **Author**

**Page 17: [5] Deleted**                                      **Author**

**Page 17: [5] Deleted**                                      **Author**

**Page 17: [6] Formatted**                                    **Author**

Font: Not Italic

**Page 17: [6] Formatted**                                    **Author**

Font: Not Italic

**Page 17: [7] Deleted**                                      **Author**

**Page 17: [7] Deleted**            **Author**

**Page 17: [7] Deleted**            **Author**

**Page 17: [7] Deleted**            **Author**

**Page 17: [8] Formatted**            **Author**

Font: Not Italic

**Page 17: [9] Formatted**            **Author**

Font: Not Bold

**Page 17: [10] Formatted**            **Author**

Font color: Auto

**Page 17: [10] Formatted**            **Author**

Font color: Auto

**Page 17: [11] Deleted**            **Author**

**Page 17: [11] Deleted**            **Author**

**Page 17: [12] Formatted**            **Author**

Font: Not Bold

**Page 17: [13] Deleted**            **Author**

**Page 17: [14] Formatted**            **Author**

Left, Indent: Left: 0 cm, Hanging: 0.75 cm

**Page 17: [15] Deleted**            **Author**

**Page 17: [15] Deleted**            **Author**

**Page 17: [15] Deleted**            **Author**

**Page 17: [15] Deleted**            **Author**

**Page 17: [16] Formatted**            **Author**

Font: Not Bold

**Page 17: [17] Deleted**            **Author**

**Page 17: [17] Deleted**          **Author**

**Page 17: [18] Deleted**          **Author**

**Page 17: [18] Deleted**          **Author**

**Page 17: [19] Formatted**          **Author**

Font: Not Italic

**Page 17: [19] Formatted**          **Author**

Font: Not Italic

**Page 17: [20] Formatted**          **Author**

Font color: Auto

**Page 17: [20] Formatted**          **Author**

Font color: Auto

**Page 17: [21] Formatted**          **Author**

Font: Not Italic

**Page 17: [21] Formatted**          **Author**

Font: Not Italic

**Page 17: [22] Deleted**          **Author**

**Page 17: [22] Deleted**          **Author**

**Page 17: [22] Deleted**          **Author**

**Page 17: [22] Deleted**          **Author**

**Page 17: [22] Deleted**          **Author**

**Page 17: [22] Deleted**          **Author**

**Page 17: [22] Deleted**          **Author**

**Page 17: [22] Deleted**          **Author**

**Page 17: [23] Formatted**          **Author**

Font: Not Italic

Font: Not Italic

**Page 17: [24] Deleted**           **Author**

**Page 17: [24] Deleted**           **Author**

**Page 17: [24] Deleted**           **Author**

**Page 17: [25] Formatted**           **Author**

Font: Not Italic

**Page 17: [25] Formatted**           **Author**

Font: Not Italic

**Page 17: [26] Deleted**           **Author**

**Page 17: [26] Deleted**           **Author**

**Page 17: [26] Deleted**           **Author**

**Page 17: [27] Deleted**           **Author**

**Page 17: [27] Deleted**           **Author**

**Page 17: [28] Formatted**           **Author**

Font: Not Bold

**Page 17: [29] Formatted**           **Author**

Font color: Auto

**Page 17: [29] Formatted**           **Author**

Font color: Auto

**Page 17: [30] Formatted**           **Author**

Font: Not Italic

**Page 17: [30] Formatted**           **Author**

Font: Not Italic

**Page 17: [31] Formatted**           **Author**

Font color: Auto

**Page 18: [32] Formatted**           **Author**

Font color: Auto

**Page 18: [33] Formatted**                             **Author**

Font: Not Italic

**Page 18: [34] Formatted**                             **Author**

Font: Not Bold

**Page 18: [35] Deleted**                             **Author**

**Page 18: [36] Formatted**                             **Author**

Left, Indent: Left:  0 cm, Hanging:  0.75 cm

**Page 18: [37] Deleted**                             **Author**

**Page 18: [37] Deleted**                             **Author**

**Page 18: [37] Deleted**                             **Author**

**Page 18: [38] Formatted**                             **Author**

Font: Not Italic

**Page 18: [38] Formatted**                             **Author**

Font: Not Italic

**Page 18: [39] Deleted**                             **Author**

**Page 18: [39] Deleted**                             **Author**

**Page 18: [39] Deleted**                             **Author**

**Page 18: [39] Deleted**                             **Author**

**Page 18: [40] Formatted**                             **Author**

Font: Not Italic

**Page 18: [40] Formatted**                             **Author**

Font: Not Italic

**Page 18: [41] Deleted**                             **Author**

**Page 18: [41] Deleted**                             **Author**

**Page 18: [41] Deleted**                                **Author**

**Page 18: [41] Deleted**                                **Author**

**Page 18: [42] Formatted**                            **Author**

Font: Not Bold

**Page 18: [43] Formatted**                            **Author**

Font color: Auto

**Page 18: [43] Formatted**                            **Author**

Font color: Auto

**Page 18: [44] Deleted**                                **Author**

**Page 18: [44] Deleted**                                **Author**

**Page 18: [45] Formatted**                            **Author**

Font: Not Bold

**Page 18: [46] Deleted**                                **Author**

**Page 18: [46] Deleted**                                **Author**

**Page 18: [46] Deleted**                                **Author**

**Page 18: [47] Formatted**                            **Author**

Font: Not Italic

**Page 18: [47] Formatted**                            **Author**

Font: Not Italic

**Page 18: [48] Deleted**                                **Author**

**Page 18: [48] Deleted**                                **Author**

**Page 18: [48] Deleted**                                **Author**

**Page 18: [49] Formatted**                            **Author**

Font color: Auto

**Page 18: [49] Formatted**                            **Author**

**Page 18: [50] Deleted**                 **Author**

**Page 18: [51] Formatted**                **Author**

Font: Not Italic

**Page 18: [51] Formatted**                **Author**

Font: Not Italic

**Page 18: [52] Deleted**                 **Author**

**Page 18: [52] Deleted**                 **Author**

**Page 18: [52] Deleted**                 **Author**

**Page 18: [52] Deleted**                 **Author**

**Page 18: [52] Deleted**                 **Author**

**Page 18: [52] Deleted**                 **Author**

**Page 18: [52] Deleted**                 **Author**

**Page 18: [53] Formatted**                **Author**

Font: Not Bold

**Page 18: [54] Deleted**                 **Author**

**Page 18: [54] Deleted**                 **Author**

**Page 18: [54] Deleted**                 **Author**

**Page 18: [54] Deleted**                 **Author**

**Page 18: [54] Deleted**                 **Author**

**Page 18: [54] Deleted**                 **Author**

**Page 18: [54] Deleted**                 **Author**

**Page 18: [54] Deleted**            **Author**

**Page 18: [55] Formatted**            **Author**

Font: Not Italic

**Page 18: [55] Formatted**            **Author**

Font: Not Italic

**Page 18: [56] Deleted**            **Author**

**Page 18: [56] Deleted**            **Author**

**Page 18: [56] Deleted**            **Author**

**Page 18: [56] Deleted**            **Author**

**Page 18: [57] Deleted**            **Author**

**Page 18: [57] Deleted**            **Author**

**Page 18: [57] Deleted**            **Author**

**Page 18: [57] Deleted**            **Author**

**Page 18: [57] Deleted**            **Author**

**Page 18: [57] Deleted**            **Author**

**Page 18: [57] Deleted**            **Author**

**Page 18: [57] Deleted**            **Author**

**Page 18: [58] Formatted**            **Author**

Font: Not Italic

**Page 18: [58] Formatted**            **Author**

Font: Not Italic

**Page 18: [59] Deleted**            **Author**

**Page 18: [59] Deleted**            **Author**

**Page 18: [60] Deleted**        **Author**

**Page 18: [60] Deleted**        **Author**

**Page 18: [60] Deleted**        **Author**

**Page 18: [60] Deleted**        **Author**

**Page 18: [61] Deleted**        **Author**

**Page 18: [61] Deleted**        **Author**

**Page 18: [62] Formatted**        **Author**

Font: Not Bold

**Page 18: [63] Formatted**        **Author**

Font color: Auto

**Page 18: [63] Formatted**        **Author**

Font color: Auto

**Page 18: [64] Deleted**        **Author**

**Page 18: [64] Deleted**        **Author**

**Page 18: [65] Formatted**        **Author**

Font: Not Italic

**Page 18: [66] Deleted**        **Author**

**Page 18: [66] Deleted**        **Author**

**Page 18: [66] Deleted**        **Author**

**Page 18: [66] Deleted**        **Author**

**Page 18: [66] Deleted**        **Author**

**Page 18: [67] Formatted**        **Author**

Font: Not Bold

**Page 18: [68] Deleted**                       **Author**

**Page 18: [68] Deleted**                       **Author**

**Page 18: [69] Formatted**                  **Author**

Font color: Auto

**Page 18: [69] Formatted**                  **Author**

Font color: Auto

**Page 18: [70] Formatted**                  **Author**

Font: Not Italic

**Page 18: [70] Formatted**                  **Author**

Font: Not Italic

**Page 19: [71] Deleted**                       **Author**

**Page 19: [71] Deleted**                       **Author**

**Page 19: [72] Formatted**                  **Author**

Font: Not Bold

**Page 19: [73] Deleted**                       **Author**

**Page 19: [73] Deleted**                       **Author**

**Page 19: [73] Deleted**                       **Author**

**Page 19: [74] Formatted**                  **Author**

Font: Not Italic

**Page 19: [75] Deleted**                       **Author**

**Page 19: [75] Deleted**                       **Author**

**Page 19: [75] Deleted**                       **Author**

**Page 19: [76] Deleted**                       **Author**

**Page 19: [76] Deleted**                       **Author**

**Page 19: [77] Formatted**          **Author**

Font: Not Bold

**Page 19: [78] Deleted**          **Author**

**Page 19: [78] Deleted**          **Author**

**Page 19: [78] Deleted**          **Author**

**Page 19: [79] Formatted**          **Author**

Font: Not Bold

**Page 19: [80] Deleted**          **Author**

**Page 19: [80] Deleted**          **Author**

**Page 19: [80] Deleted**          **Author**

**Page 19: [80] Deleted**          **Author**

**Page 19: [80] Deleted**          **Author**

**Page 19: [80] Deleted**          **Author**

**Page 19: [81] Formatted**          **Author**

Font: Not Italic

**Page 19: [81] Formatted**          **Author**

Font: Not Italic

**Page 19: [82] Formatted**          **Author**

Font: Not Bold

**Page 19: [83] Deleted**          **Author**

**Page 19: [83] Deleted**          **Author**

**Page 19: [84] Formatted**          **Author**

Font: Not Bold

**Page 19: [85] Deleted**          **Author**

**Page 19: [85] Deleted**                                       **Author**

**Page 19: [86] Formatted**                                **Author**

Font: Not Italic

**Page 19: [86] Formatted**                                **Author**

Font: Not Italic

**Page 19: [87] Deleted**                                      **Author**

**Page 19: [87] Deleted**                                      **Author**

**Page 19: [87] Deleted**                                      **Author**

**Page 19: [87] Deleted**                                      **Author**

**Page 19: [87] Deleted**                                      **Author**

**Page 19: [87] Deleted**                                      **Author**

**Page 19: [88] Formatted**                                **Author**

Font: Not Italic

**Page 19: [89] Formatted**                                **Author**

Font: Not Bold

**Page 19: [90] Formatted**                                **Author**

Left

**Page 19: [91] Formatted**                                **Author**

Left, Indent: Left:  0 cm, Hanging:  0.75 cm

**Page 19: [92] Deleted**                                      **Author**

**Page 19: [92] Deleted**                                      **Author**

**Page 19: [93] Formatted**                                **Author**

Font: Not Italic

**Page 19: [94] Deleted**                                      **Author**

**Page 19: [95] Deleted**          **Author**

**Page 19: [95] Deleted**          **Author**

**Page 19: [95] Deleted**          **Author**

**Page 19: [95] Deleted**          **Author**

**Page 19: [96] Formatted**          **Author**

Font color: Auto

**Page 19: [96] Formatted**          **Author**

Font color: Auto

**Page 19: [97] Deleted**          **Author**

**Page 19: [97] Deleted**          **Author**

**Page 19: [98] Formatted**          **Author**

Font: Not Italic

**Page 19: [98] Formatted**          **Author**

Font: Not Italic

**Page 19: [99] Formatted**          **Author**

Font color: Auto

**Page 19: [99] Formatted**          **Author**

Font color: Auto

**Page 19: [100] Formatted**          **Author**

Font: Not Italic

**Page 19: [100] Formatted**          **Author**

Font: Not Italic

**Page 19: [101] Deleted**          **Author**

**Page 19: [101] Deleted**          **Author**

**Page 19: [102] Formatted**          **Author**

Font: Not Italic

| Page 19: [103] Deleted | Author |
|---|---|

| Page 19: [103] Deleted | Author |
|---|---|

| Page 19: [103] Deleted | Author |
|---|---|

| Page 19: [103] Deleted | Author |
|---|---|

| Page 19: [103] Deleted | Author |
|---|---|

| Page 19: [104] Formatted | Author |
|---|---|

Font: Not Bold

| Page 19: [105] Deleted | Author |
|---|---|

| Page 19: [105] Deleted | Author |
|---|---|

| Page 19: [106] Formatted | Author |
|---|---|

Font: Not Italic

| Page 19: [106] Formatted | Author |
|---|---|

Font: Not Italic

| Page 19: [107] Formatted | Author |
|---|---|

Font color: Auto

| Page 19: [107] Formatted | Author |
|---|---|

Font color: Auto

| Page 19: [108] Formatted | Author |
|---|---|

Font: Not Italic

| Page 19: [108] Formatted | Author |
|---|---|

Font: Not Italic

| Page 19: [109] Deleted | Author |
|---|---|

| Page 19: [109] Deleted | Author |
|---|---|

| Page 19: [109] Deleted | Author |
|---|---|

| Page 19: [110] Formatted | Author |
|---|---|

Font: Not Italic

| Page 19: [111] Deleted | Author |
|---|---|

| Page 19: [111] Deleted | Author |
|---|---|

| Page 19: [111] Deleted | Author |
|---|---|

| Page 19: [112] Formatted | Author |
|---|---|

Font color: Auto

| Page 19: [112] Formatted | Author |
|---|---|

Font color: Auto

| Page 19: [113] Formatted | Author |
|---|---|

Font: Not Italic

| Page 19: [113] Formatted | Author |
|---|---|

Font: Not Italic

| Page 20: [114] Deleted | Author |
|---|---|

| Page 20: [114] Deleted | Author |
|---|---|

| Page 20: [114] Deleted | Author |
|---|---|

| Page 20: [114] Deleted | Author |
|---|---|

| Page 20: [114] Deleted | Author |
|---|---|

| Page 20: [114] Deleted | Author |
|---|---|

| Page 20: [114] Deleted | Author |
|---|---|

| Page 20: [115] Formatted | Author |
|---|---|

Font: Not Italic

| Page 20: [115] Formatted | Author |
|---|---|

Font: Not Italic

| Page 20: [116] Deleted | Author |
|---|---|

**Page 20: [116] Deleted**          **Author**

**Page 20: [116] Deleted**          **Author**

**Page 20: [116] Deleted**          **Author**

**Page 20: [117] Deleted**          **Author**

**Page 20: [117] Deleted**          **Author**

**Page 20: [117] Deleted**          **Author**

**Page 20: [118] Deleted**          **Author**

**Page 20: [118] Deleted**          **Author**

**Page 20: [119] Formatted**          **Author**

Font: Not Italic

**Page 20: [119] Formatted**          **Author**

Font: Not Italic

**Page 20: [120] Deleted**          **Author**

**Page 20: [120] Deleted**          **Author**

**Page 20: [120] Deleted**          **Author**

**Page 20: [120] Deleted**          **Author**

**Page 20: [121] Deleted**          **Author**

**Page 20: [121] Deleted**          **Author**

**Page 20: [121] Deleted**          **Author**

**Page 20: [121] Deleted**          **Author**

**Page 20: [121] Deleted**          **Author**

**Page 20: [121] Deleted**          **Author**

**Page 20: [121] Deleted**          **Author**

**Page 20: [121] Deleted**          **Author**

**Page 20: [122] Formatted**          **Author**

Font: Not Italic

**Page 20: [122] Formatted**          **Author**

Font: Not Italic

**Page 20: [123] Deleted**          **Author**

**Page 20: [123] Deleted**          **Author**

**Page 20: [123] Deleted**          **Author**

**Page 20: [123] Deleted**          **Author**

**Page 20: [123] Deleted**          **Author**

**Page 20: [123] Deleted**          **Author**

**Page 20: [123] Deleted**          **Author**

**Page 20: [124] Formatted**          **Author**

Font: Not Italic

**Page 20: [124] Formatted**          **Author**

Font: Not Italic

**Page 20: [125] Deleted**          **Author**

**Page 20: [125] Deleted**          **Author**

**Page 20: [125] Deleted**          **Author**

**Page 20: [125] Deleted**          **Author**

**Page 20: [125] Deleted**          **Author**

**Page 20: [125] Deleted**          **Author**

**Page 20: [125] Deleted**          **Author**

**Page 20: [125] Deleted**          **Author**

**Page 20: [125] Deleted**          **Author**

**Page 20: [125] Deleted**          **Author**

**Page 20: [126] Formatted**          **Author**

Font: Not Italic

**Page 20: [126] Formatted**          **Author**

Font: Not Italic

**Page 20: [127] Deleted**          **Author**

**Page 20: [127] Deleted**          **Author**

**Page 20: [127] Deleted**          **Author**

**Page 20: [127] Deleted**          **Author**

**Page 20: [127] Deleted**          **Author**

**Page 20: [127] Deleted**          **Author**

**Page 20: [127] Deleted**          **Author**

**Page 20: [128] Formatted**          **Author**

Font: Not Italic

**Page 20: [128] Formatted**          **Author**

Font: Not Italic

**The relevance of mid-Holocene Arctic warming to the future**

**(Supplementary table and figures)**

Masakazu Yoshimori[1,2*] and Marina Suzuki[3]

[1] Faculty of Environmental Earth Science, Global Institution for Collaborative Research and Education, and Arctic Research Center,
Hokkaido University, Sapporo, Japan

[2] Atmosphere and Ocean Research Institute, The University of Tokyo, Kashiwa, Japan

[3] Graduate School of Environmental Science, Hokkaido University, Sapporo, Japan

Manuscript is to be submitted to *Climate of the Past* (22 June 2019)

* Corresponding author address: Atmosphere and Ocean Research Institute, The University of Tokyo, 5-1-5, Kashiwanoha, Kashiwa, Chiba 277-8568 Japan
Phone: +81-4-7136-4380, Fax: +81-4-7136-4375, E-mail: masakazu@aori.u-tokyo.ac.jp

1    Table S1    Model years used to construct the long-term climatology for the PI and MH simulations from

2                          "r1i1p1" runs in the CMIP5 dataset.

| Model | PI | MH |
|---|---|---|
| bcc-csm1-1 | 0001 - 0500 | 0001 - 0100 |
| CCSM4 | 0800 - 1300 | 1000 - 1300 |
| CNRM-CM5 | 1850 - 2699 | 1950 - 2149 |
| CSIRO-Mk3-6-0 | 0001 - 0500 | 0001 - 0100 |
| FGOALS-g2 | 0001 - 0900 | 0340 - 1024 |
| FGOALS-s2 | 1850 - 2350 | 0001 - 0100 |
| GISS-E2-R | 3331 - 4530 | 2500 - 2599 |
| IPSL-CM5A-LR | 2370 - 2799 | 2701 - 2800 |
| MIROC-ESM | 1800 - 2429 | 2330 - 2429 |
| MRI-CGCM3 | 1851 - 2350 | 1951 - 2050 |

[Figure]

8    Fig. S1    Seasonal progress of the zonal mean radiative forcing calculated with the insolation anomaly for

9    ΔMH and planetary albedo from the PI experiment (W m$^{-2}$). The mean of all 10 models was used. See main

10    text for details.

[Figure]

14  Fig. S2   Surface air temperature anomaly (°C) for ΔMH from the reconstruction (left) and simulations

15  (right): (a) & (b) annual mean, (c) & (d) warmest month, and (e) & (f) coldest month. The reconstruction

16  data are taken from the extended data of Bartlein et al. (2011). The mean of all 10 model simulations was

17  used.

[Figure]

20
21    Figure S3    Simulated and diagnosed surface temperature changes (°C) for the land (north of 60°N): (a)
22    ΔMH; and (b) ΔRCP4.5. The black polygonal solid lines denote simulated changes and blue polygonal
23    dashed lines denote the sum of diagnosed partial changes; two lines are superimposed. The graphs represent
24    the means of all 10 models listed in Table 2. See Table 3 for the interpretation of each component.

[Figure]

28    Figure S4   Fractional contribution of individual processes to the simulated surface temperature change

29    (%) over the land (north of 60°N) for ΔMH and ΔRCP4.5: (a) spring (March-April-May); (b) summer (June-

30    July-August); (c) autumn (September-October-November); and (d) winter (December-January-February)

31    means. The sum of the bar graphs in the same color for each plot adds up to 100%. The hatching indicates

32    the contribution is statistically significant at the 10% level. All 10 models listed in Table 2 are used. See

33    Table 3 for the interpretation of each component. Note that the vertical scale for (b) is three times larger

34    than others.

**Reference**

Bartlein, P. J., Harrison, S. P., Brewer, S., Connor, S., Davis, B. A. S., Gajewski, K., Guiot, J., Harrison-Prentice, T. I., Henderson, A., Peyron, O., Prentice, I. C., Scholze, M., Seppa, H., Shuman, B., Sugita, S., Thompson, R. S., Viau, A. E., Williams, J., and Wu, H.: Pollen-based continental climate reconstructions at 6 and 21 ka: a global synthesis. Clim. Dyn., 37, 775-802, https://doi.org/10.1007/s00382-010-0904-1, 2011.

**Page 7: [1] Deleted**                                **Author**

**Page 7: [1] Deleted**                                **Author**